# Proteomic mapping of cytosol-facing outer mitochondrial and ER membranes in living human cells by proximity biotinylation

Victoria Hung[1†‡], Stephanie S Lam[1†§], Namrata D Udeshi[2], Tanya Svinkina[2], Gaelen Guzman[2], Vamsi K Mootha[2,3], Steven A Carr[2], Alice Y Ting[1,2#*]

[1]Department of Chemistry, Massachusetts Institute of Technology, Cambridge, United States; [2]Broad Institute of MIT and Harvard, Cambridge, United States; [3]Department of Molecular Biology, Howard Hughes Medical Institute, Massachusetts General Hospital, Harvard Medical School, Boston, United States

*For correspondence:
ayting@stanford.edu

†These authors contributed equally to this work

Present address: ‡Departments of Developmental Biology and Genetics, Stanford University, Stanford, United States; §Department of Molecular Biology, Howard Hughes Medical Institute, Massachusetts General Hospital, Harvard Medical School, Boston, United States; #Departments of Biology, Genetics, and Chemistry, Stanford University, Stanford, United States

**Abstract** The cytosol-facing membranes of cellular organelles contain proteins that enable signal transduction, regulation of morphology and trafficking, protein import and export, and other specialized processes. Discovery of these proteins by traditional biochemical fractionation can be plagued with contaminants and loss of key components. Using peroxidase-mediated proximity biotinylation, we captured and identified endogenous proteins on the outer mitochondrial membrane (OMM) and endoplasmic reticulum membrane (ERM) of living human fibroblasts. The proteomes of 137 and 634 proteins, respectively, are highly specific and highlight 94 potentially novel mitochondrial or ER proteins. Dataset intersection identified protein candidates potentially localized to mitochondria-ER contact sites. We found that one candidate, the tail-anchored, PDZ-domain-containing OMM protein SYNJ2BP, dramatically increases mitochondrial contacts with rough ER when overexpressed. Immunoprecipitation-mass spectrometry identified ribosome-binding protein 1 (RRBP1) as SYNJ2BP's ERM binding partner. Our results highlight the power of proximity biotinylation to yield insights into the molecular composition and function of intracellular membranes.

DOI: https://doi.org/10.7554/eLife.24463.001

## Introduction

The mitochondrion and endoplasmic reticulum (ER) are two centrally important cellular organelles, and their cytosol-facing membranes engage in a wide variety of essential processes, including calcium exchange, cell death, immune signaling, lipid biosynthesis, and protein translation, secretion, and import. High quality proteomic maps of each of these membranes—the outer mitochondrial membrane (OMM) and the ER membrane (ERM)—would be extremely valuable for clarifying the sub-organellar location of known mitochondrial and ER-resident proteins, and likely identify novel proteins not previously associated with either organelle. Yet such maps remain elusive. Each organelle has been mapped in its entirety via mass spectrometry (MS)-based proteomics multiple times (*Calvo and Mootha, 2010*; *Chen et al., 2010*). However, even proteomes such as MitoCarta 2.0 (*Calvo et al., 2016*), which appear close to complete with respect to mitochondrial matrix, inner membrane, and intermembrane space (IMS) proteins, under-represent OMM proteins, likely because the outer membrane is depleted during purification of the organelle. OMM material can be specifically enriched for MS analysis through osmotic shock lysis of the outer membrane of purified mitochondria followed by sucrose gradient ultracentrifugation (*Niemann et al., 2013*; *Schmitt et al.,*

*2006*; *Zahedi et al., 2006*), but this fractionation protocol is highly disruptive and imperfect, which compromises the quality of the resulting datasets. Furthermore, mammalian OMMs have never previously been mapped. For whole-ER datasets, the propensity of ER-derived microsomes to co-centrifuge with virtually all other cellular compartments (*Sadowski et al., 2008*) has rendered these proteomes, which include both ERM and ER lumen contents, similarly unclean with poor overlap between independent datasets (*Chen et al., 2010*). Rapoport and coworkers were able to obtain a highly specific list of 25 abundant ERM proteins via microsome enrichment from dog pancreas (*Shibata et al., 2010*), but these represent only a miniscule fraction of the proteins thought to reside at mammalian ER membranes.

Because OMM and ER membranes are so difficult to purify to homogeneity, and existing enrichment protocols also lose large fractions of resident proteins, these membranes are ideal candidates for an alternative proteomic approach known as proximity biotinylation. In this approach (*Figure 1A*), which bypasses biochemical fractionation and organelle purification altogether, a 27 kDa engineered monomeric peroxidase called APEX2 (*Lam et al., 2015*) is genetically targeted to the cellular region of interest. Addition of hydrogen peroxide ($H_2O_2$) and a membrane-permeant substrate for APEX2, biotin-phenol (BP), results in the covalent biotinylation of endogenous proteins within a few nanometers of APEX2. Because labeling is performed over a one-minute time window while cells and organelles are intact, spatial relationships between proteins are preserved, resulting in their specific tagging by the biotin probe (*Hung et al., 2014*; *Loh et al., 2016*; *Rhee et al., 2013*). After labeling, cells are lysed, and biotinylated proteomes are enriched with streptavidin beads and analyzed by MS.

Importantly, we have shown that the APEX-generated biotin-phenoxyl radical does not cross cellular membranes (*Rhee et al., 2013*). Therefore, in this study, the proteomic mapping should be specific for proteins on the outer leaflet of the ER membrane and not tag proteins inside the ER lumen. The OMM contains porins that allow free passage of molecules <5 kDa, including the BP radical. Hence, we expect some biotinylation of IMS-resident proteins and some IMM (inner mitochondrial membrane) proteins with IMS exposure. However, no BP radical should enter the mitochondrial matrix (*Figure 1A*).

Here, we describe our efforts to map the proteomes of the OMM and ERM in human embryonic kidney (HEK) 293T cells via stable expression of APEX2 fusion constructs and BP labeling for one minute. We used a ratiometric strategy employing Stable Isotope Labeling by Amino acids in Cell culture (SILAC) (*Hung et al., 2014*) to quantify the relative proximity of each MS-detected protein to the OMM or ERM versus cytosol. Together, the two proteomes identified 94 potentially novel mitochondrially- or ER-localized proteins, one of which we validated by fluorescence microscopy.

Additionally, we intersected our OMM and ERM lists in an attempt to identify candidate proteins that reside at mitochondria-ER contact sites. Mitochondria-ER contacts have long been observed by electron microscopy (EM) (*Copeland and Dalton, 1959*; *Csordás et al., 2006*; *Meier et al., 1981*; *Shore and Tata, 1977*), and patches of ER membrane co-purify with mitochondria (the mitochondria-associated membrane, or MAM) (*Vance, 1990*). More recent work has shown that mitochondria-ER contact sites are functionally important for calcium homeostasis (*Rizzuto et al., 1998*; *Szabadkai et al., 2006*), lipid biosynthesis (*Kornmann et al., 2011*, *Kornmann et al., 2009*; *Lewin et al., 2002*; *Rusiñol et al., 1994*), cellular apoptosis (*Iwasawa et al., 2011*), and regulation of mitochondrial fission (*Friedman et al., 2011*; *Murley et al., 2013*). Kornmann et al. used a genetic screen in yeast to discover the ERMES (ER-Mitochondria Encounter Structure) complex, an ER-mitochondrial tether in yeast (*Kornmann et al., 2009*). However, mammalian cells do not have homologs to the ERMES complex. More generally, only a handful of proteins localized to mitochondria-ER contact sites have been identified with confidence in any cell type (*Paillusson et al., 2016*). A high-quality inventory of the protein components within this important subcellular region would advance the field of mammalian mitochondria-ER contact biology.

By mining our OMM and ERM proteomic data, we discovered that the tail-anchored OMM protein synaptojanin-2 binding protein SYNJ2BP (also known as OMP25 (*Nemoto and De Camilli, 1999*)) was highly enriched by *both* OMM- and ERM-targeted APEX2. Follow-up experiments showed that overexpression of SYNJ2BP in HEK 293T cells leads to a dramatic increase in mitochondrial contacts specifically with rough ER membrane, mediated by SYNJ2BP's binding partner on the ER membrane, RRBP1.

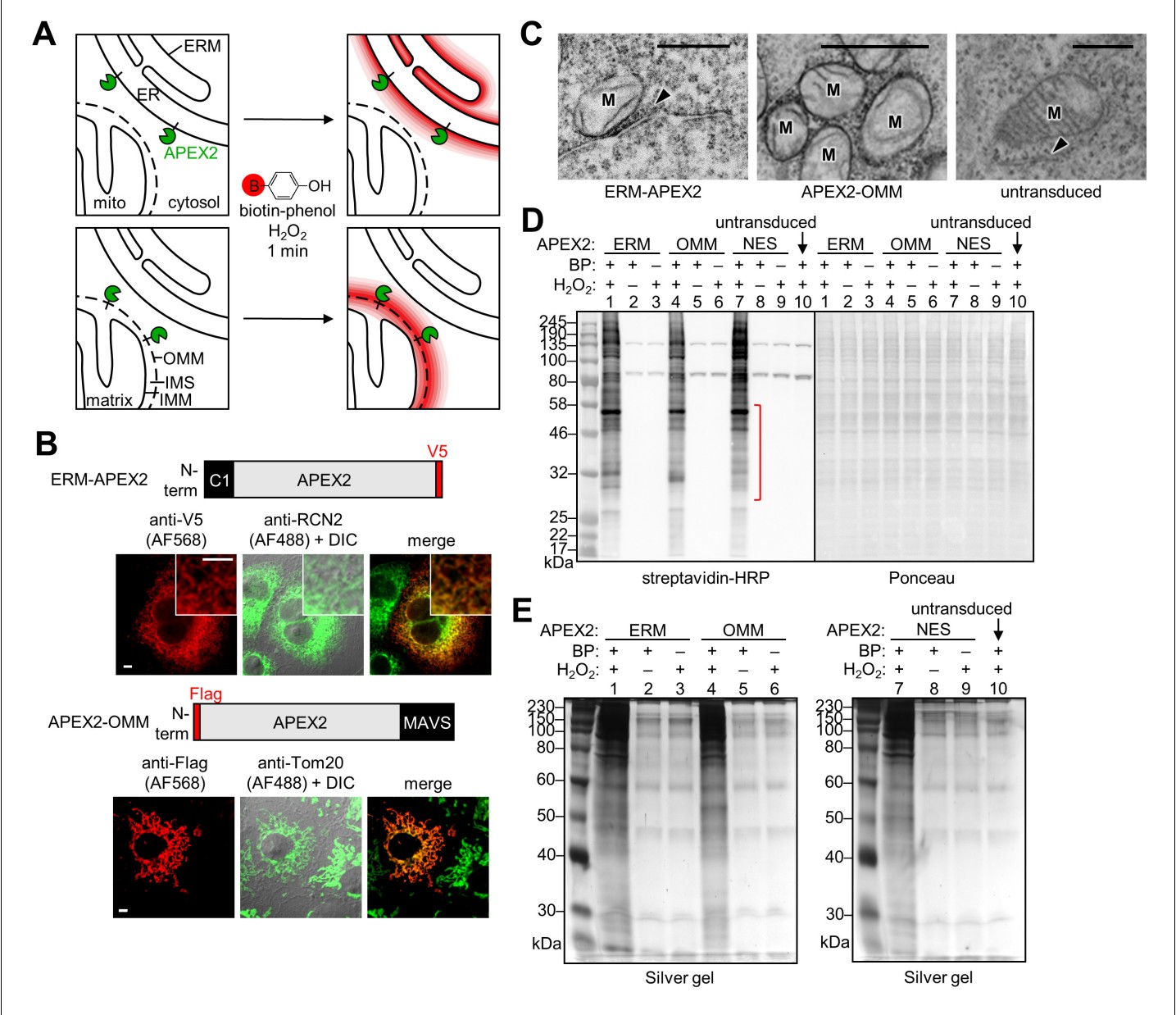

**Figure 1.** APEX2 proteomic labeling scheme and characterization of APEX2 fusion constructs on the ER and outer mitochondrial membranes. (**A**) Scheme of APEX2-catalyzed biotinylation at the endoplasmic reticulum membrane (ERM, top) and outer mitochondrial membrane (OMM, bottom). APEX2 (green) was targeted to the ERM facing the cytosol, via fusion to the N-terminal 27 amino acids of the ER-resident protein P450 oxidase 2C1 and to the OMM via fusion to the C-terminal 31 amino acids of mitochondrial antiviral-signaling protein MAVS. Live cells were pre-incubated with biotin-phenol (red B = biotin) for 30 min, and hydrogen peroxide ($H_2O_2$) was added to initiate the biotinylation reaction for 1 min. Cells were quenched and then lysed, and the biotinylated proteins were enriched with streptavidin beads and analyzed by mass spectrometry. IMS, intermembrane space. IMM, inner mitochondrial membrane. (**B**) Confocal fluorescence imaging of ERM-APEX2 and APEX2-OMM fusion constructs. Constructs were introduced into COS-7 cells using lentivirus. Two days after transduction, the cells were fixed and stained with anti-V5 to detect ERM-APEX2 (Alexa Fluor 568 (AF568) readout), or anti-Flag to detect APEX2-OMM (AF568 readout). Anti-RCN2 was used to visualize the ER, or anti-Tom20 to visualize mitochondria (AF488 readout). Fluorescence channels are not normalized. Scale bars, 5 μm. The insets show a zoomed in portion of the cell. DIC, differential interference contrast. (**C**) Electron microscopy (EM) characterization of ERM-APEX2 and APEX2-OMM localization. ERM-APEX2 and APEX2-OMM genes were introduced into human embryonic kidney (HEK) 293T cells using lentivirus. Cells were fixed and overlaid with a solution of diaminobenzidine and $H_2O_2$. APEX2 catalyzes the localized polymerization of diaminobenzidine, which then reacts with osmium to provide contrast for EM (***Lam et al., 2015***; ***Martell et al., 2012***). Dark regions indicate APEX2 activity. ER indicated by arrowheads. M, mitochondria. Untransduced HEK cell shown for comparison. Scale bars, 500 nm. The untransduced and APEX2-OMM images in this figure overlap with data shown in Supplementary Figure 7 of ***Lam et al. (2015)***. (**D**) Analysis of ERM-APEX2 and APEX2-OMM-catalyzed biotinylation by streptavidin blot. HEK 293T cells stably expressing ERM-

*Figure 1 continued on next page*

*Figure 1 continued*

APEX2, APEX2-OMM, APEX2-NES (NES = nuclear export sequence), or no APEX2 construct (lane 10) were labeled live as in (**A**). Whole cell lysates were separated by gel electrophoresis and blotted with streptavidin-horseradish peroxidase (streptavidin-HRP) on the left, or stained with Ponceau on the right. BP, biotin-phenol. The red bracket indicates a molecular weight range where the banding patterns produced by these APEX2 constructs differ notably. (**E**) Analysis of streptavidin-enriched biotinylated proteomes by silver stain. ERM-APEX2, APEX2-OMM, and APEX2-NES stable cells were labeled live as in (**A**), lysed, and incubated with streptavidin beads to enrich for biotinylated proteins. Eluted proteins were separated by gel electrophoresis and visualized by silver stain.

DOI: https://doi.org/10.7554/eLife.24463.002

The following figure supplement is available for figure 1:

**Figure supplement 1.** Characterization of ERM-APEX2 and APEX2-OMM-catalyzed biotinylation by imaging.

DOI: https://doi.org/10.7554/eLife.24463.003

## Results

### Targeting APEX2 to the OMM and ERM and characterization of biotin labeling

To target APEX2, we fused the gene to 31- and 27-amino acid targeting domains of the native OMM and ERM proteins MAVS (*Seth et al., 2005*) and cytochrome P450 2C1 (*Ahn et al., 1993*), respectively (*Figure 1B*). Correct localization was confirmed by fluorescence microscopy in conjunction with mitochondrial and ER markers (*Figure 1B*). For higher-resolution analysis, we capitalized on the ability of APEX2 to generate contrast for electron microscopy (EM) (*Lam et al., 2015*; *Martell et al., 2012*). After cell fixation, targeted APEX2 catalyzed the oxidative polymerization and local deposition of 3,3'-diaminobenzidine (DAB) in the presence of $H_2O_2$. The DAB polymer recruits electron-dense osmium, which provides contrast for EM. *Figure 1C* shows that APEX2-OMM stains the outer membrane of mitochondria, while ERM-APEX2 darkly stains the ERM. The DAB/osmium density of ERM-APEX2 spreads into the cytosol but is excluded from the ER lumen, confirming that APEX2 faces the cytosol rather than the ER lumen, as designed.

Because APEX2 has a weak dimerization tendency (*Lam et al., 2015*), high expression levels on organellar membranes can give rise to undesired aggregation or morphological perturbation (*Lam et al., 2015*; *Snapp et al., 2003*). To avoid these, we generated HEK 293T cells stably expressing APEX2-OMM or ERM-APEX2 at low levels following lentiviral infection. To check for adequate biotinylation activity under these low expression conditions, we pre-incubated APEX2 cells with biotin-phenol for 30 min and added $H_2O_2$ for 1 min, then fixed and stained with neutrAvidin-fluorophore conjugate to visualize biotinylated proteins. *Figure 1—figure supplement 1* shows that each sample displayed robust biotinylation, as visualized by neutrAvidin staining, compared to a negative control with APEX2 omitted. Next, we repeated the live cell biotinylation, lysed the cells, and analyzed the whole cell lysates by streptavidin blotting (*Figure 1D*). Each APEX2 fusion biotinylates a wide range of endogenous proteins, and encouragingly, the banding patterns are distinguishable from one another and from that of APEX2-NES, a soluble whole-cytosol APEX2 control (NES is a nuclear export sequence).

### Proteomic mapping of the OMM and ERM

Because the OMM and ERM are 'open' subcellular domains continuous with the cytosol and lacking membrane enclosures to trap the biotin-phenoxyl radical, we employed our previously described 'ratiometric SILAC' approach (*Hung et al., 2016*, *Hung et al., 2014*) to ensure high spatial specificity in proteomic mapping. In this approach, each protein is quantified not only by its extent of biotinylation by targeted APEX2 (APEX2-OMM or ERM-APEX2), but also by its extent of biotinylation by a reference APEX2 construct, in this case cytosolic APEX2-NES. Thus, a cytosolic, non-OMM protein such as kinesin may be weakly biotinylated by APEX2-OMM, but if it is biotinylated to a greater extent by the more proximal APEX2-NES, it will be filtered out of the final OMM proteome.

Before generating proteomic samples, we tested our streptavidin enrichment conditions (*Figure 1E*). Lysates from cells biotinylated with APEX2-OMM, ERM-APEX2, or APEX2-NES were incubated with streptavidin-coated magnetic beads and then subjected to a series of denaturing washes intended to remove all proteins except those directly covalently biotinylated by APEX2. After boiling in SDS to elute from beads, the biotinylated proteomes were run on SDS-PAGE and

visualized by silver stain. *Figure 1E* shows that our protocol enriches much more protein from experimental samples than from negative controls with APEX2, $H_2O_2$, or BP omitted. The residual bands in these negative controls likely represent the endogenous biotinylated proteins present in all mammalian cells (*Chapman-Smith and Cronan, 1999*) in addition to some non-specific bead binders not removed by the washes.

We generated proteomic samples for ratiometric SILAC analysis according to the scheme in *Figure 2A*. Two replicates were performed for the OMM, and two for the ERM. Each replicate consisted of three cellular samples: one cultured with heavy isotope-labeled arginine and lysine ('H') and biotinylated with APEX2-OMM or ERM-APEX2; one cultured in medium-isotope arginine and lysine ('M') and biotinylated with the reference construct APEX2-NES; and one negative control sample cultured in light amino acids ('L') with either APEX2 or $H_2O_2$ omitted. After separate treatment with biotin-phenol followed by quenching and cell lysis, the H, M, and L lysates of each replicate were pooled, and the mixture was subjected to streptavidin bead enrichment as in *Figure 1E*. Eluted proteins were run on SDS-PAGE to reduce sample complexity, and 16 gel bands were excised and individually digested with trypsin and analyzed by liquid chromatography-tandem MS (LC-MS/MS).

Due to our experimental design, each MS-detected protein was associated with three quantified MS peaks—H, M, and L—reflecting its respective enrichment in the APEX2-OMM or ERM-APEX2 sample, in the cytosolic APEX2 reference sample, and in the negative control. We used SILAC ratios, which normalize for variations in absolute protein abundance across replicates, to quantify parameters of interest: $\log_2$(H/L) for the extent of protein biotinylation by APEX2-OMM or ERM-APEX2; $\log_2$(M/L) for the extent of protein biotinylation by APEX2-NES; and most importantly, $\log_2$(H/M) for the *relative* biotinylation of any given protein by OMM- or ERM-targeted APEX2 versus cytosolic APEX2-NES, which is a measure of protein proximity to the OMM or ERM versus the open cytosol.

The $\log_2$(H/L) and $\log_2$(H/M) values were well-correlated across independent replicates ($R^2 \geq 0.74$) (*Figure 2—figure supplement 1A*). Histograms of $\log_2$(H/L) values showed broad, right-shifted distributions (*Figure 2—figure supplement 1B and C*). To check for enrichment of known mitochondrial and ER proteins, we plotted in green the subset of detected proteins known to be ER- or mitochondrially-localized, and we plotted in red the subset of detected proteins expected to be false positives. The green (true positive) populations were clearly right-shifted compared to the red (false positive) populations, indicating appropriate enrichments in both $\log_2$(H/L) and $\log_2$(H/M) for both OMM and ERM replicates (*Figure 2—figure supplement 1D and E*).

To define OMM and ERM proteomes, we calculated SILAC ratio cut-offs, above which we retained proteins (because they are likely true positives) and below which we removed proteins (because they are likely false positives). To calculate the cut-offs, we made use of prior knowledge—proteins previously identified in the literature as bona fide mitochondrial, ER, non-mitochondrial, or non-ER proteins—and we generated receiver operating characteristic (ROC) curves (*Figure 2—figure supplement 1F*). We selected SILAC ratios (both $\log_2$(H/L) and $\log_2$(H/M)) that maximized the retention of known true positives and minimized the retention of known false positives. These cut-offs are shown as dashed lines in both the histograms in *Figure 2—figure supplement 1D and E* and in the two-dimensional scatter plots in *Figure 2B*. From the scatter plots, we were encouraged to see that known true positives (green) cluster in the top right quadrant due to both high $\log_2$(H/M) and high $\log_2$(H/L) ratios, while proteins that lack prior mitochondrial or ER annotation largely cluster below the $\log_2$(H/M) cut-off.

After application of both $\log_2$(H/L) and $\log_2$(H/M) cut-offs and intersection of the two replicates, we obtained a final proteome of 137 proteins for the OMM (*Supplementary file 1a*) and a final proteome of 634 proteins for the ERM (*Supplementary file 1b*).

## Analysis of OMM and ERM proteomes

With our OMM and ERM proteomes defined, we proceeded to characterize the quality of each dataset in terms of specificity and sensitivity (depth-of-coverage). First, we analyzed the mitochondrial specificity of the OMM proteome. In the entire human proteome, 8% of proteins have mitochondrial annotation according to MitoCarta (*Pagliarini et al., 2008*), the Gene Ontology Cell Component (GOCC) database (*Ashburner et al., 2000*), our APEX-mapped mitochondrial matrix (*Rhee et al., 2013*) or IMS (*Hung et al., 2014*) proteomes, or literature. In our OMM proteome, 84% have prior mitochondrial annotation (*Figure 2C*). The remaining 16% of proteins could be either false positives or 'mitochondrial orphans' (*Supplementary file 1c*)—true positives not previously known to be

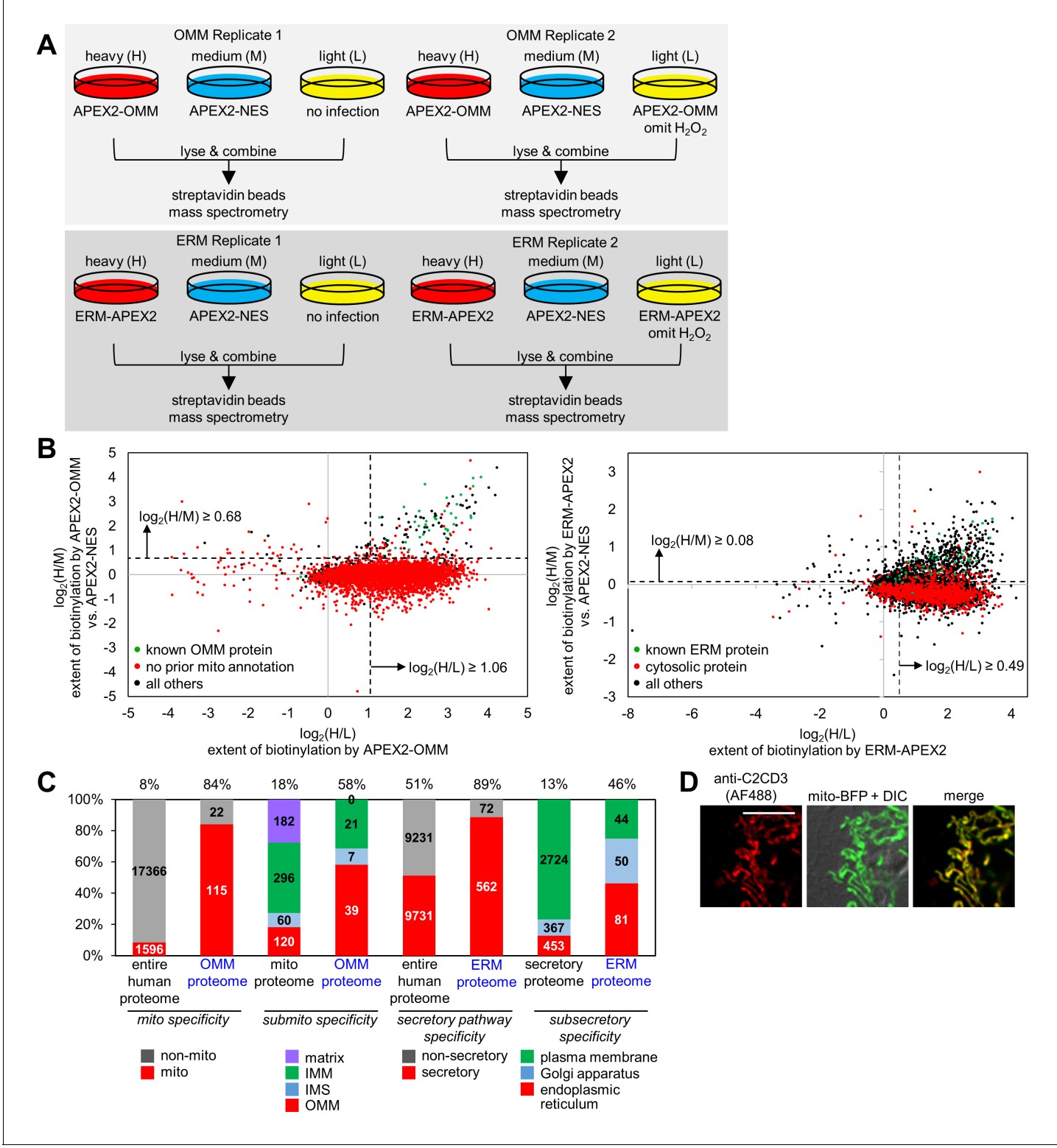

**Figure 2.** OMM and ERM SILAC proteomic experiments in HEK 293T cells. (**A**) Design of two 3-state SILAC (Stable Isotope Labeling by Amino acids in in Cell culture) experiments. HEK 293T cells stably expressing APEX2-OMM or ERM-APEX2 were cultured in media containing heavy (H) isotopes of arginine and lysine. APEX2-NES stable cells were cultured in media with medium (M) isotopes of arginine and lysine, and negative control cells where either expression of APEX2 was omitted or ERM-APEX2 or APEX2-OMM stable cells were not treated with $H_2O_2$ were cultured in light (L) media. All dishes were then identically treated with biotin-phenol and $H_2O_2$ except for the negative controls mentioned before. The cells were separately lysed,

*Figure 2 continued on next page*

*Figure 2 continued*

and the three lysates in each replicate were combined 1:1:1 by protein mass. The mixture was then enriched using streptavidin beads and analyzed by mass spectrometry. (B) Scatter plot showing $\log_2$(H/M) versus $\log_2$(H/L) for Replicate 1 for the OMM proteomic experiment (left) and the ERM proteomic experiment (right). For each of the 5296 proteins detected in Replicate 1 of the OMM proteomic experiment, the $\log_2$(H/M) value was plotted against the $\log_2$(H/L) value. Known OMM proteins (members of our OMM true positive list, *Supplementary file 2a*) are shown in green; proteins without prior mitochondrial annotation according to MitoCarta (*Pagliarini et al., 2008*), Gene Ontology Cell Component (GOCC) (*Ashburner et al., 2000*) annotation, or the APEX-mapped mitochondrial matrix (*Rhee et al., 2013*) or mitochondrial intermembrane space (IMS) (*Hung et al., 2014*) proteomes are shown in red; all other proteins are shown in black. SILAC ratio cut-offs used to filter the OMM mass spectrometric data and obtain the final OMM proteome are shown by the dashed lines. On the right, a similar analysis was conducted for the 4799 proteins detected in Replicate 1 of the ERM proteomic experiment. Known ERM proteins (from our ERM true positive list, *Supplementary file 2b*) are shown in green; proteins with cytosolic annotation are shown in red; all others are shown in black. (C) Characterization of the specificity of the OMM and ERM proteomes. The first two columns show the percentage of proteins in the entire human proteome and our OMM proteome with prior mitochondrial annotation according to MitoCarta (*Pagliarini et al., 2008*), GOCC (*Ashburner et al., 2000*), and our APEX-mapped mitochondrial matrix (*Rhee et al., 2013*) and mitochondrial IMS (*Hung et al., 2014*) proteomes. The next two columns show the breakdown of proteins with GOCC sub-mitochondrial annotation available (mitochondrial matrix, IMM, IMS, or OMM). If a protein had more than one sub-mitochondrial annotation, we assigned it to one of the four categories in this priority: OMM>IMS>IMM>matrix. Columns 5 and 6 show the percentage of proteins with secretory pathway annotation according to GOCC (*Ashburner et al., 2000*) or Phobius (*Käll et al., 2004*, *Käll et al., 2007*) in the entire human proteome and our ERM proteome, respectively. The last two columns show the breakdown of proteins with GOCC sub-secretory annotation available. Protein assignments were based on the following priority: endoplasmic reticulum>Golgi apparatus>plasma membrane. (D) Fluorescence imaging of endogenous C2CD3, a mitochondrial orphan, in COS-7 cells. COS-7 cells were transfected with mitochondrially-targeted blue fluorescent protein (mito-BFP) and mCherry-KDEL (not shown). The cells were fixed and stained with an antibody against endogenous C2CD3. Scale bar, 10 µm.

DOI: https://doi.org/10.7554/eLife.24463.004

The following figure supplement is available for figure 2:

**Figure supplement 1.** Analysis of the OMM and ERM proteomic data to obtain the final proteomes.

DOI: https://doi.org/10.7554/eLife.24463.005

mitochondrial but identified as such by our study. Several of these 22 proteins have ER annotation and thus could be ER proteins that interact with mitochondria. For example, CANX and TMX1 have both been enriched in MAM preparations (*Lynes et al., 2012*). We obtained an antibody against one of the mitochondrial orphans, C2CD3, and confirmed by immunofluorescence staining that it is indeed localized to mitochondria (*Figure 2D*). This result has interesting implications, as C2CD3 is associated with the centriole during interphase and mitosis and is required for recruitment of distal appendage proteins (*Ye et al., 2014*); our result hints at a possible link between mitochondria and cilium formation.

We next characterized the sub-mitochondrial specificity of the OMM proteome. Of the known human mitochondrial proteins with OMM, IMS, IMM, or matrix sub-mitochondrial annotation according to GOCC (*Supplementary file 2c*) (*Ashburner et al., 2000*), 18% have OMM annotation. By the same method of analysis, 58% of our OMM proteome has prior OMM annotation, indicating that we have enriched OMM proteins. As expected, our dataset also includes some well-known IMS and IMM proteins (e.g. OXPHOS-related subunits NDUFA8, NDUFB1, NDUFB6, NDUFB7, NDUFB10, NDUFB11, CYC1, COX6C, COX4l1, SCO1/2, CYCS) because biotin-phenoxyl radicals generated by APEX2-OMM can pass through porins in the OMM and enter the IMS. Consistent with the impermeability of the IMM to biotin-phenoxyl radicals, we did not enrich any mitochondrial matrix-resident proteins.

To characterize the specificity of our ERM proteome, we calculated the fraction of proteins with prior secretory pathway annotation, according to GOCC (*Ashburner et al., 2000*), Phobius (*Käll et al., 2004*, *Käll et al., 2007*), and literature. This value is 89%, compared to only 51% for the entire human proteome (*Figure 2C*). We then characterized the sub-secretory specificity of the ERM proteome. Of known human secretory proteins with ER, Golgi apparatus, or plasma membrane annotation according to GOCC (*Supplementary file 2d*) (*Ashburner et al., 2000*), 13% have ER annotation; by the same metric, 46% of our ERM proteome has ER annotation, which indicates that we have enriched for ER-resident proteins over Golgi apparatus and plasma membrane proteins. Complete exclusion of these latter proteins is not expected, as many are translated on or near the ER membrane before trafficking to their final destination.

Finally, we inspected our ERM list for soluble proteins known to reside in the ER lumen and detected only 13 (*Supplementary file 2e*). Previous studies showed that two of these proteins,

ERO1L (*Cabibbo et al., 2000*) and HSP90B1 (*Reddy et al., 1999*), have protease-accessible regions on the outer surface of ER-derived microsomes. This suggests that they actually possess regions topologically facing the cytosol, consistent with our data, and inconsistent with their current GOCC annotations. The other 11 ER lumen proteins we enriched may have previously-unrecognized cytosolic exposure, or perhaps they are biotinylated by ERM-APEX2 during retro-translocation to the cytosol upon ERAD (ER-associated protein degradation) (*Tsai et al., 2002*).

The coverage of our OMM proteome was estimated by first generating a true positive list of 79 well-established OMM proteins by literature curation (*Supplementary file 2a*). We found that over half (53%) of these proteins were present in our OMM proteome. Analogously, the coverage of the ERM proteome was calculated using a curated list of 90 established ERM proteins (*Supplementary file 2b*), resulting in 44% coverage. As discussed previously (*Hung et al., 2014*, *Hung et al., 2016*), factors known to limit coverage may also be at play here, including failure to biotinylate proteins that are sterically shielded in the live cell context, and removal of proteins that are dual-labeled (*Supplementary files 1f and 1g*), i.e. strongly biotinylated by *both* targeted APEX2 and cytosolic APEX2, resulting in $\log_2$(H/M) values that do not pass the $\log_2$(H/M) cut-offs shown in the bottom right quadrants of the scatter plots in *Figure 2B*.

In summary, like previous proteomes mapped with APEX, the OMM and ERM proteomes obtained in this study feature high specificity paired with moderate coverage and should be valuable resources for biologists studying signaling processes on mitochondrial and ER membranes.

## Overlap between OMM and ERM proteomes

Our focus on ER and mitochondrial membranes provides a unique opportunity to mine for insights into contact regions between these organelles. For MS proteomic analysis, the only existing method to enrich mitochondria-ER contact sites is to purify MAMs by subjecting crude mitochondria to an additional round of centrifugation. Loosely associated ER-derived microsomes separate into a distinct layer—the MAM layer—apart from purified mitochondria, and this layer is extracted for proteomic analysis. Several MAM proteomic studies have generated lists up to 1212 proteins long (*Poston et al., 2013*). Most of these are populated with contaminants, including proteins known to localize to the mitochondrial matrix or ER lumen.

We sought to mine our OMM and ERM proteomes, which were obtained without biochemical fractionation, for novel proteins that might reside at mitochondria-ER contact sites. Intersecting the 137 OMM and 634 ERM proteins, we found that 68 proteins were enriched by both APEX2-OMM and ERM-APEX2 (*Supplementary file 1e*). While many of these could be dual-localized to both mitochondrial and ER membranes (e.g. ACSL1 (*Lewin et al., 2001*; *Milger et al., 2006*), MARCH5 (*Nakamura et al., 2006*; *Sugiura et al., 2013*; *Yonashiro et al., 2006*), and ARMC10 (*Huang et al., 2003*; *López-Doménech et al., 2012*)), some proteins could be specific residents of mitochondria-ER contact sites. Encouragingly, our intersected list includes MFN2, PTPIP51 (also known as FAM82A2), MFF, TMX1, and ATAD3A—proteins linked to mammalian mitochondria-ER contact sites by previous studies (*de Brito and Scorrano, 2008*; *Friedman et al., 2011*; *Issop et al., 2015*; *Krols et al., 2016*; *Stoica et al., 2014*). DRP1, a predominantly cytosolic protein that localizes to mitochondria-ER contact sites during mitochondrial fission (*Friedman et al., 2011*), did not give enough unique MS-detected peptides to enter our datasets.

## An overexpression screen identifies SYNJ2BP

To identify proteins in our intersected list with a potential role in tethering the ERM to the OMM, analogous to the ERMES complex in yeast, we developed a gain-of-function assay. Proteins with a role in tethering might increase the extent of mitochondria-ER overlap upon overexpression. Hence, we obtained V5 epitope-tagged plasmids for 40 proteins in our list, and for the 38 genes with correct sequences, we overexpressed them via Lipofectamine 2000 transfection in COS-7 cells. We used fluorescence microscopy to check for increased overlap between mitochondria and ER markers in these cells (*Supplementary file 1e*). Only a single protein in our screen produced such a phenotype, synaptojanin-2 binding protein, or SYNJ2BP. Interestingly, SYNJ2BP also had the highest combined enrichment across both OMM and ERM proteomes (*Supplementary file 1e*). As shown in *Figure 3A*, low expression of SYNJ2BP-V5 produces cells with distinct and well-separated ER and mitochondrial morphologies, whereas high expression causes mitochondria and ER to re-organize

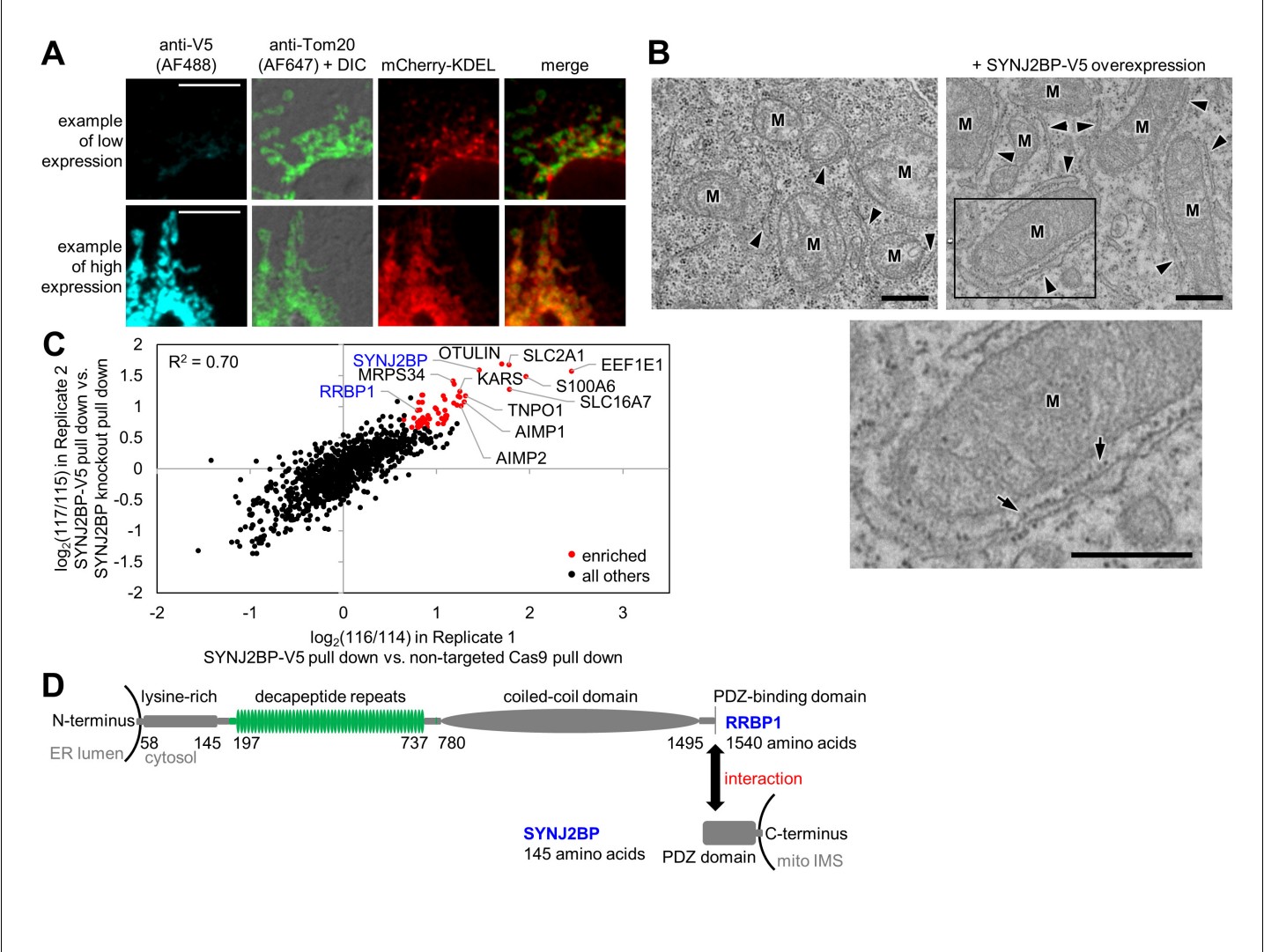

**Figure 3.** Identification of SYNJ2BP and RRBP1 as a potential mitochondria-ER tethering complex. (A) SYNJ2BP overexpression assay in COS-7 cells. COS-7 cells were transfected with SYNJ2BP-V5 and mCherry-KDEL (an ER marker) using Lipofectamine 2000. ~26 hr later, cells were fixed and stained with anti-V5 to detect SYNJ2BP and anti-Tom20 to visualize mitochondria. Based on anti-V5 signal intensity, cells with high or low SYNJ2BP expression were identified and imaged within the heterogeneous population. Scale bars, 10 μm. (B) EM characterization of HEK 293T cells overexpressing SYNJ2BP-V5. HEK 293T were transfected with 1 μg V5-APEX2-NLS (NLS = nuclear localization signal) alone, *left*, or with 2 μg SYNJ2BP-V5 and 1 μg V5-APEX2-NLS, using Lipofectamine 2000. The cells were fixed, stained with DAB and osmium, and then processed for EM. Micrographs are shown for cells with positive nuclear staining, indicating transfection (see zoomed out views in *Figure 3—figure supplement 1B*). 'M', mitochondria. Arrowheads point to ER. Bottom image shows zoom of the boxed region. Arrows point to ribosomes. Scale bars, 500 nm. (C) Correlation between replicates of the SYNJ2BP-V5 immunoprecipitation-mass spectrometry (IP-MS) experiment. To identify binding partners of SYNJ2BP, a population of HeLa cells stably expressing SYNJ2BP-V5 with endogenous SYNJ2BP knocked out was generated. SYNJ2BP-V5 was enriched from whole cell lysates using anti-V5 antibody coupled to protein G beads. The scatter plot shows the correlation between two independent IP-MS replicate experiments. Proteins deemed significantly enriched (by a moderated *t*-test; p-value<0.02) are colored in red and all others in black. (D) Structural domains of SYNJ2BP and RRBP1. For RRBP1 (top), regions in green vary among isoforms; isoforms vary in the number of decapeptide repeats (up to 54 repeats) and in the presence or absence of three amino acids after the decapeptide repeat region. The C-terminus of RRBP1 contains a consensus PDZ-binding domain (S/T-X-Φ-COOH, where X is any amino acid and Φ is a hydrophobic amino acid, usually V/I/L; TSV in RRBP1). SYNJ2BP is a tail-anchored OMM protein with a cytosolic PDZ domain.

DOI: https://doi.org/10.7554/eLife.24463.006

The following figure supplement is available for figure 3:

**Figure supplement 1.** Characterization of SYNJ2BP and identification of RRBP1 as a binding partner.

DOI: https://doi.org/10.7554/eLife.24463.007

and strongly co-localize. We obtained the same results using a SYNJ2BP overexpression plasmid lacking any epitope tag, showing that the observed phenotype is tag-independent (data not shown). To examine this effect at higher spatial resolution, we repeated the assay using EM, which more clearly visualizes the spatial relationships between ER and mitochondrial membranes. *Figure 3B* and *Figure 3—figure supplement 1B* show that SYNJ2BP-V5 overexpression causes ER membranes to dramatically 'zip up' and form extended contacts along mitochondrial surfaces. Quantitation across ≥7 fields of view per condition showed that the median percentage of mitochondrial surface in contact with ER increased from 6% to 25% per mitochondrion upon SYNJ2BP-V5 overexpression. We were also very intrigued to find that the induced mitochondria-ER contacts were highly uniform in width (~45 nm spacing between mitochondria and ER membranes), and nearly all were filled with ribosomes.

SYNJ2BP is a 16 kDa tail-anchored OMM protein with a cytosol-facing PDZ domain. PDZ domains are almost exclusively found in soluble proteins that help anchor signaling components at the plasma membrane (*Fanning and Anderson, 1999*; *Romero et al., 2011*)—for example, PSD-95, which anchors synaptic receptors to the cytoskeleton. The presence of a PDZ domain in a transmembrane mitochondrial protein is highly unusual and suggestive of a binding partner. While no previous study has pinpointed a mitochondrial function for SYNJ2BP, multiple interaction partners have been identified, including synaptojanin 2A, an inositol 5'-phosphatase (*Nemoto and De Camilli, 1999*), low density liproprotein receptor-related protein (LRP), and megalin (*Gotthardt et al., 2000*). SYNJ2BP has also been reported as a negative regulator of angiogenesis (*Adam et al., 2013*) and an inhibitor of tumor growth and metastasis (*Liu et al., 2016*). A splice variant of SYNJ2BP in mice with a different C-terminus, ARIP2, was reported to localize to the cytoplasm and bind ACTRIIA and ACTRIIB, although it was never addressed whether ARIP2 was mitochondrial (*Matsuzaki et al., 2002*).

By immunofluorescence staining, endogenous SYNJ2BP protein localizes throughout the mitochondrion and does not appear enriched at sites of ER overlap (*Figure 3—figure supplement 1A*). If SYNJ2BP plays a role in mitochondria-ER tethering under physiological conditions (without overexpression) perhaps only a subset of the protein (e.g., a post-translationally modified form) engages with the ER membrane.

## Discovery of RRBP1 as a binding partner for SYNJ2BP

Having identified SYNJ2BP as a protein with a possible role in mitochondria-ER tethering, we sought to identify its binding partner on the ERM. To do this, we used CRISPR (clustered regularly interspaced short palindromic repeats)/Cas9 (*Ran et al., 2013*) to remove endogenous SYNJ2BP and selected for HeLa cells stably expressing V5-tagged SYNJ2BP as a replacement (*Figure 3—figure supplement 1C*). We performed two replicate immunoprecipitations of SYNJ2BP-V5 with anti-V5 coated beads alongside negative control cells lacking SYNJ2BP-V5. Samples were separately digested to peptides, labeled with iTRAQ reagents (isobaric Tags for Relative and Absolute Quantification (*Ross et al., 2004*)), mixed together, and analyzed by LC-MS/MS (*Figure 3—figure supplement 1D*). The scatter plot in *Figure 3C* shows the 56 proteins (colored red) that were significantly enriched (moderated *t*-test false discovery rate of p-value<0.02) in the experimental sample over the negative control in both replicates (*Supplementary file 3a*). As expected, we enriched the known SYNJ2BP binding partner E-SYT1 (*Christianson et al., 2012*). Strikingly, we also enriched the entire MARS (multi aminoacyl tRNA synthetase) complex of 11 proteins (*Lee et al., 2004*), and other proteins related to protein translation (the signal recognition particle receptor subunit beta SRPRB, dolichyl-diphosphooligosaccharide–protein glycosyltransferase subunit STT3A, ribophorin 1 RPN1, eIF-2-alpha kinase activator GCN1L1, and mitochondrial ribosomal proteins MRPS34 and MRPS27).

We searched the list of 56 SYNJ2BP interactors for an ER-localized, PDZ-binding motif-containing protein that might serve as a tethering partner for SYNJ2BP. There was only one such protein, the ribosome-binding protein 1 (RRBP1), a 166 kDa single-pass transmembrane ER protein that was also enriched in our ERM proteome. The C-terminal cytosolic domain of RRBP1, which constitutes 98% of the entire protein, contains a lysine-rich region, up to 54 decapeptide repeats, a coiled-coil domain, and a PDZ-binding domain (consensus sequence TSV) at its C-terminal end (*Figure 3D*). RRBP1 was originally proposed to be a ribosome receptor (*Wanker et al., 1995*), but more recent work has identified mRNA binding (*Cui et al., 2012*) and microtubule binding and bundling (*Ogawa-Goto et al., 2007*) functions. RRBP1 has also been reported to bind to kinesin (*Diefenbach et al., 2004*). Shibata et al. found that RRBP1 is enriched in the sheet-like ER (*Shibata et al., 2010*).

To validate the interaction between SYNJ2BP and RRBP1, we performed immunoprecipitation on HEK 293T cells transiently overexpressing SYNJ2BP-V5 and blotted for endogenous RRBP1 in the V5-enriched material. *Figure 4A* shows that anti-V5 immunoprecipitation enriches RRBP1, which appears as multiple bands due to its various splice isoforms. We were unable to perform the reverse immunoprecipitation due to the lack of antibodies suitable for immunoprecipitating endogenous RRBP1 and the intractability of cloning recombinant RRBP1 with its many repeats.

To determine if RRBP1 is the relevant binding partner for mediating overexpressed SYNJ2BP's ability to increase mitochondria-ER contacts, we repeated our gain-of-function assay in RRBP1 CRISPR-mediated knockout cells. *Figure 4—figure supplement 1A* shows that endogenous RRBP1 protein is indeed absent from these cells. By EM, the induced mitochondria-ER contacts we previously observed upon SYNJ2BP overexpression were absent. In parallel, a positive control with SYNJ2BP overexpressed in wild-type cells (without knockout of endogenous RRBP1) showed the same robust induction of mitochondria-ER contacts (*Figure 4B* and *Figure 4—figure supplement 1B*) as before (*Figure 3B*). Therefore, RRBP1 is essential for SYNJ2BP's unique gain-of-function phenotype, perhaps because the two proteins bind across mitochondria-ER junctions, via a probable PDZ domain-PDZ binding motif interaction (*Figure 3D*).

Finally, we performed imaging of endogenous RRBP1, stained in COS-7 cells with anti-RRBP1 antibody. RRBP1 localization extended across the ER, without any clear enrichment at sites of mitochondria-ER overlap (*Figure 4—figure supplement 1C*). If RRBP1 plays a role in mitochondria-ER tethering under physiological, non-overexpression conditions, perhaps only a subset of the protein (a specific splice variant or post-translationally modified form of RRBP1) localizes to mitochondria-ER contact sites.

## A link to protein translation

Due to RRBP1's reported functions in ribosome and/or mRNA binding, the striking appearance of ribosomes in mitochondria-ER contacts induced by SYNJ2BP overexpression (*Figure 3B*), and the numerous translation-related proteins enriched in our SYNJ2BP-V5 pull down (*Figure 3C*), we hypothesized that the SYNJ2BP-RRBP1 tether may be linked to protein translation. To examine this hypothesis, we repeated the immunoprecipitation/Western blot experiment after treatment of SYNJ2BP-V5-expressing live cells with protein translation inhibitors. *Figure 4A* shows that while cycloheximide treatment had no effect, puromycin reduced the extent of SYNJ2BP-V5 interaction with endogenous RRBP1 without changing the total abundance of either protein. Because puromycin is known to disassemble polysomes, while cycloheximide inhibits translation while keeping polysomes intact (*Blobel and Sabatini, 1971*; *Wettstein et al., 1964*), perhaps polysomes—a hallmark of active protein translation—regulates the extent of SYNJ2BP-RRBP1 interaction.

To examine this effect in greater detail, we repeated the assay but used EM as our readout instead of Western blot (*Figure 4C* and *Figure 4—figure supplement 2A*). In SYNJ2BP-V5-overexpressing HEK cells, we found that puromycin treatment abolished the induced mitochondria-ER contacts while cycloheximide did not. This is consistent with the results of immunoprecipitation/Western blot (*Figure 4A*). However, the high information content of EM images allowed us to perform further analysis. We separately quantified the fraction of mitochondrial membrane in contact with smooth ER or rough ER in each sample. The graphs in *Figure 4D and E* show that under conditions of SYNJ2BP-V5 overexpression, protein translation inhibitors have no effect on mitochondria-smooth ER contacts, but puromycin selectively disrupts mitochondria-rough ER contacts.

The next question that arises is whether protein translation inhibitors affect mitochondria-rough ER contacts in wild-type cells that do not overexpress SYNJ2BP-V5. We examined this, but found no significant effect (*Figure 4—figure supplement 2B*). Because mitochondria-rough ER contacts are so scarce in wild-type HEK 293T cells, and EM is not a high-throughput method, greater statistical power may be needed to observe a significant effect, if one exists.

## Analysis of SYNJ2BP and RRBP1 knockout cells

If SYNJ2BP and RRBP1 function as a mitochondria-ER tethering complex under physiological conditions, we might expect to observe a phenotype upon knockout of either protein. We used CRISPR/Cas9 to generate HEK 293T and HeLa cells missing either SYNJ2BP or RRBP1 (*Figure 4—figure supplement 1A*, *Figure 4—figure supplement 3A, and B*). Fluorescence microscopy showed no clear

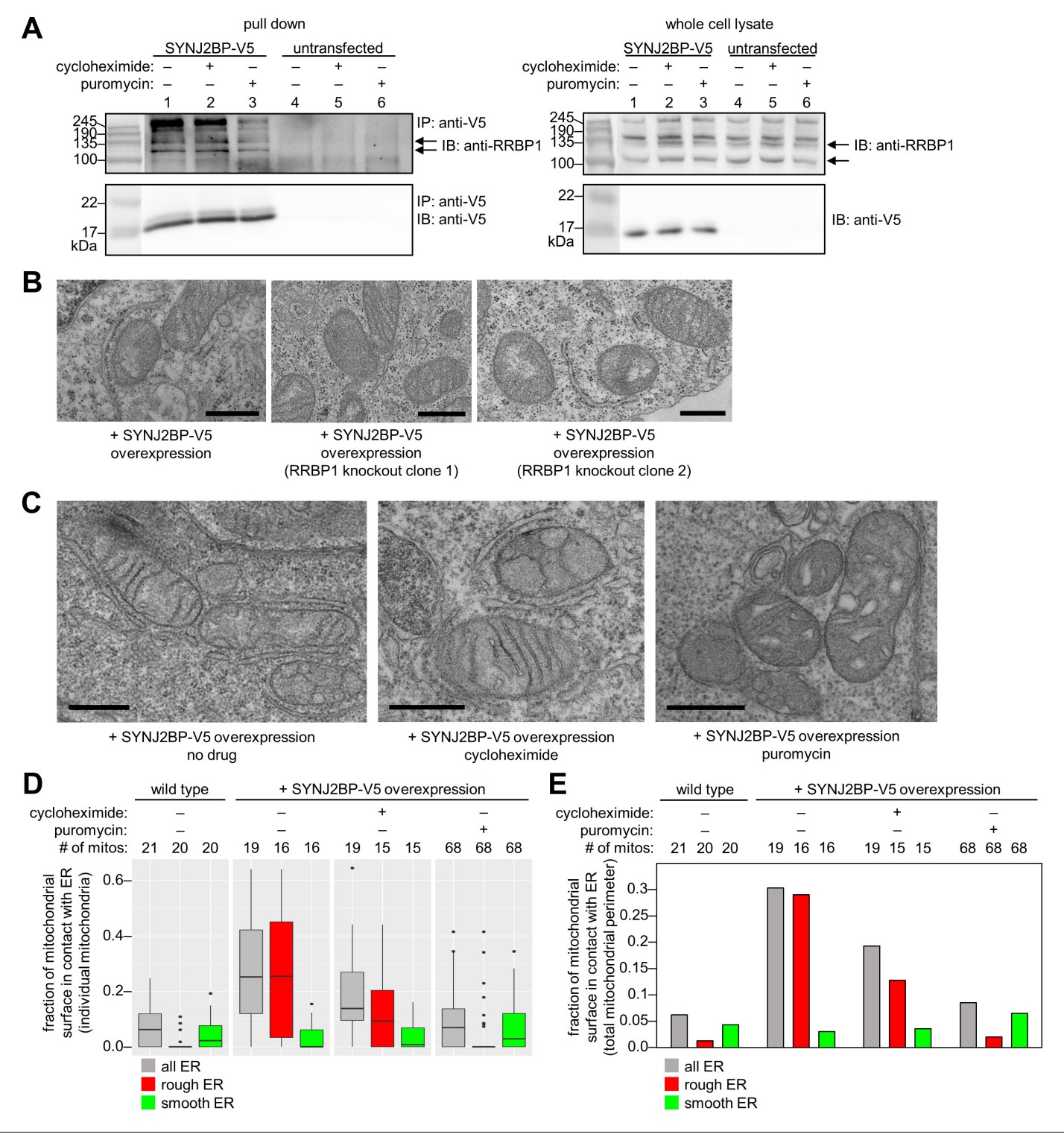

**Figure 4.** Characterization of the SYNJ2BP-RRBP1 interaction. (**A**) Pull down of SYNJ2BP-V5 from cells treated with protein translation inhibitors. HEK 293T cells expressing SYNJ2BP-V5 (Lipofectamine 2000 transfection) were treated with 200 μM cycloheximide or puromycin for 2 hr before cell lysis. Lysates were enriched with anti-V5 coated beads and blotted for endogenous RRBP1 (top) and SYNJ2BP-V5 (bottom), *left*. Lanes 4–6 show untransfected controls. *Right*, gels show same analysis of whole cell lysates prior to anti-V5 enrichment. Black arrows indicate RRBP1 bands. (**B**) EM of HEK 293T cells overexpressing SYNJ2BP-V5 in the presence versus absence of RRBP1 knockout (KO). SYNJ2BP-V5 and Flag-APEX2-NLS (a nuclear marker) were transfected into wild-type cells, *left,* and two different RRBP1 KO clonal cell lines, *middle* and *right*. After 24 hr, the cells were fixed, stained with DAB and osmium, and processed for EM. Micrographs are shown for cells with positive nuclear stain (indicating transfection). Additional

*Figure 4 continued on next page*

*Figure 4 continued*

fields of view in *Figure 4—figure supplement 1B*. Scale bars, 500 nm. (C) EM of HEK 293T cells overexpressing SYNJ2BP-V5 and treated with protein translation inhibitors as in (A). The cells were fixed and stained as in (B). Scale bars, 500 nm. Additional EM fields of view in *Figure 4—figure supplement 2A*. (D) Quantitation of EM images in (C) and *Figure 4—figure supplement 2A*. Box plots showing the percent of mitochondrial surface in contact (<80 nm) with the ER. Untransfected cells also analyzed for comparison (left). (E) Same as D, but a single percentage is shown for each condition that reflects the total length of mitochondrial perimeter in contact with ER (total, rough, or smooth), divided by the total length of mitochondrial perimeter measured across ≥6 fields of view per condition (15–68 total mitochondria per condition).
DOI: https://doi.org/10.7554/eLife.24463.008

The following source data and figure supplements are available for figure 4:

**Source data 1.** Spreadsheet containing the mitochondrial perimeter length, type of ER in contact with each mitochondrion, mitochondrion-ER contact length, and average distance between the mitochondrion and the ER for each mitochondrion analyzed in *Figure 4D and E*.
DOI: https://doi.org/10.7554/eLife.24463.011
**Figure supplement 1.** Additional characterization of the SYNJ2BP-RRBP1 complex and of RRBP1.
DOI: https://doi.org/10.7554/eLife.24463.009
**Figure supplement 2.** Effect of protein translation inhibitors on mitochondria-ER contacts.
DOI: https://doi.org/10.7554/eLife.24463.010
**Figure supplement 3.** Functional characterization of SYNJ2BP and RRBP1 knockout (KO) cells.
DOI: https://doi.org/10.7554/eLife.24463.012

reduction in mitochondria-ER overlap in these cells (*Figure 4—figure supplement 3A and B*). Due to the connection to protein translation observed in the experiments above, we also assayed for differences in cell growth rates (*Figure 4—figure supplement 3C*), preferential utilization of glucose versus galactose as a carbon source (*Figure 4—figure supplement 3D*), and protein synthesis rates (*Figure 4—figure supplement 3E*). None of these assays produced significant differences between the knockout cell lines and wild-type HEK 293T or HeLa cells.

## Discussion

Our study provides two high-quality proteomic maps for two subcellular regions that are highly important but have been incompletely studied. The OMM is crucial for mitochondrial fusion and fission (*van der Bliek et al., 2013*), import of mitochondrial proteins (*Neupert, 1997*), interaction with cytoskeleton and motor proteins (*Sheng, 2014*), cellular apoptosis (*Tait and Green, 2010*), innate immune signaling (*Seth et al., 2005*), and communication with other organelles, including the ER (*Rowland and Voeltz, 2012*), nucleus (*Kotiadis et al., 2014*), and peroxisomes (*Thoms et al., 2009*). The ERM mediates calcium signaling, lipid synthesis, protein translation, secretion, and folding, as well as protein export and degradation under certain conditions (*Schwarz and Blower, 2016*). Our APEX2-derived proteomic maps (*Supplementary files 1a and 1b*) contain most of these protein classes, and enrich several more (22 in the OMM proteome (*Supplementary file 1c*), and 72 in the ERM proteome (*Supplementary file 1d*)) that were not previously known to reside at these membranes. Hence, the lists are valuable resources for furthering our understanding of the OMM and ERM and for generating novel hypotheses.

Mitochondria and ER organelles and sub-compartments have traditionally been studied by biochemical fractionation. The first step of this procedure, cell lysis, fragments the ER network into microsomes that co-sediment with many other cellular components, leading to high false discovery rates (*Croze and Morré, 1984*). By contrast, APEX2-mediated proximity biotinylation does not require membrane fractionation and tags the relevant proteomes while cells are still alive, with native membranes, structures, and protein complexes intact. Previously, we used APEX to map the proteomes of the human mitochondrial matrix (>94% mitochondrial specificity, 85% coverage (*Rhee et al., 2013*)) and IMS (>94% mitochondrial specificity, 65% coverage (*Hung et al., 2014*)). The OMM is the most open/unbounded of these mitochondrial sub-compartments and therefore is the most challenging in terms of spatial specificity. However, we again achieved high specificity (>84% specificity for mitochondrial proteins over cytosolic proteins, *Figure 2C*), attesting to the small labeling radius of APEX2-catalyzed biotinylation. As with previous studies using ratiometric SILAC (*Hung et al., 2014*) or iTRAQ (*Loh et al., 2016*) for quantitation, our coverage of known OMM proteins (*Supplementary file 2a*) was modest (53%), primarily because our workflow filters

out potentially dual-localized proteins that are comparably biotinylated by APEX2-OMM and cytosolic APEX2 (*Supplementary file 1f*). This demonstrates a significant limitation of our method, as many cytosolic factors known to play key roles in dynamic processes at the OMM or the ERM will be missed. Conversely, the spatial stringency of this methodology may enable the study of changes in protein localization to or away from the membrane in response to stimuli such as the induction of cell stress.

To demonstrate the utility of our OMM and ERM proteomic datasets to further biological discovery, we focused on mitochondria-ER contact sites, a cellular sub-domain of intense study that has been functionally linked to diverse processes. Amongst the 68 proteins that appeared in both OMM and ERM proteomes (*Supplementary file 1e*), SYNJ2BP was the most highly enriched, and displayed an intriguing phenotype in an overexpression screen (*Figure 3A and B*). Further investigation revealed that the new mitochondria-ER contacts induced by SYNJ2BP overexpression were highly uniform in width (~45 nm) and filled with ribosomes. These contacts were dependent on the presence of RRBP1 (likely to be SYNJ2BP's binding partner on the ERM) and sensitive to puromycin. While our results collectively suggest that SYNJ2BP-RRBP1 may be a novel tether that joins rough ER and mitochondrial membranes and links these contact sites to protein translation, our data are not conclusive for a number of reasons. First, all our observations were obtained under non-physiological conditions of SYNJ2BP-V5 overexpression. Second, we do not observe either protein enriched at mitochondria-ER contact sites, which is expected for a true tethering complex (*Eisenberg-Bord et al., 2016*). As suggested above, it is possible that specific isoforms or post-translationally modified variants of each protein do concentrate at contact sites, but we currently lack the tools to examine this. Third, we attempted several loss-of-function assays, using CRISPR to knock out SYNJ2BP or RRBP1, but failed to observe changes in cell growth rate, protein synthesis rate, or the extent of mitochondria-ER contacts by fluorescence microscopy (*Figure 4—figure supplement 3*). Perhaps such experiments would be more meaningful if performed in cell types with a greater quantity of native mitochondria-rough ER contacts than HEK 293T cells display (only 1.3% of the total mitochondrial perimeter is in contact with rough ER, according to quantitation in *Figure 4E*). For instance, Wang et al. used HT-1080 fibrosarcoma cancer cells with abundant mitochondria-rough ER contacts by EM to observe a decrease in these contacts upon AMFR (GP78) knockdown (*Wang et al., 2015*). Interestingly, HT-1080 cells have particularly high levels of SYNJ2BP mRNA compared to other cell types (*Wu et al., 2013*). Another possible explanation for our negative results is that redundant mitochondria-ER tethers or signaling proteins may obscure the effect of SYNJ2BP/RRBP1 knockout. The concept of 'auxiliary tethers' has been proposed (*Eisenberg-Bord et al., 2016*)—proteins that are not required for the formation of a contact site, but do still form tethers and have function.

Nevertheless, our preliminary data stemming from our ERM and OMM proteomic studies raise an intriguing hypothesis and direct focus to mitochondria-rough ER contacts specifically, which have so far received less attention than mitochondria-smooth ER contacts. Previously characterized mammalian tethers, such as IP3R-GRP75-VDAC1 (*Szabadkai et al., 2006*), PTPIP51-VAPB (*Stoica et al., 2014*), BAP31-FIS1 (*Iwasawa et al., 2011*), and MFN2 (*de Brito and Scorrano, 2008*; *Filadi et al., 2015*; *Naon et al., 2016*) appear to be specific for ribosome-excluded mitochondria-smooth ER contacts. However, mitochondria-rough ER contacts have long been observed by EM (*Csordás et al., 2006*; *Meier et al., 1981*; *Montisano et al., 1982*; *Shore and Tata, 1977*; *Giacomello and Pellegrini, 2016*; *Wang et al., 2015*). It is worth noting that historically, the term 'mitochondria-associated membranes' (MAMs) referred simply to the sub-fraction of ER microsomes that co-purify with mitochondria (*Vance, 1990*). Intriguingly, in that initial study, comparisons of SDS-PAGE profiles between MAMs and rough or smooth ER samples found a qualitatively closer resemblance between MAMs and smooth ER. Subsequent work in the field has often equated MAMs to the diverse mitochondria-ER contact sites observed by EM, but there is no direct evidence in the literature that purified MAMs encompass all mitochondria-ER contact sites including both rough and smooth ER.

It seems likely that mitochondrial contacts with rough or smooth ER create distinct subdomains with unique specialized functions, only a subset of which are captured via traditional biochemical purification. In support of this, SYNJ2BP, which we identified in our study, has not previously been enriched in any MS-based MAM proteomic study (*Horner et al., 2015*; *Liu et al., 2015*; *Poston et al., 2013*; *Zhang et al., 2011*). Indeed, the majority of functions thus far attributed to mitochondria-ER contact sites—e.g. lipid shuttling and calcium signaling—would be expected at

smooth ER contacts due to the closer membrane proximity and the specialized role of smooth ER in cholesterol metabolism and calcium storage. To date, no specific function has been assigned to mitochondria-rough ER contacts.

If SYNJ2BP-RRBP1 is a true tether that binds to ribosomes at mitochondria-rough ER contacts, can we speculate about its possible function? Perhaps these contacts are sites for translation of dual-targeted proteins destined for both mitochondria and ER, such as BCL2 (*Krajewski et al., 1993*). Alternatively, they could host the translation of mitochondrially-destined proteins, such as integral OMM or IMM proteins, that would benefit from machinery on or near the ERM. Translation activity increases the extent of SYNJ2BP-RRBP1 interaction, bringing the mitochondria closer to the rough ER, which could facilitate targeting of protein products to the mitochondria.

In conclusion, our study extends APEX2 proximity biotinylation to unbounded intracellular compartments with high spatial specificity, produces two high-quality proteomic maps of the human OMM and ERM, and demonstrates that these proteomic datasets can be mined for novel mitochondria-ER junction protein candidates.

## Materials and methods

### Plasmids and cloning

Standard restriction enzyme digest and ligation with T4 DNA ligase or Gibson cloning was used to generate all constructs. Except for those listed in the table below, the V5-tagged constructs used in our overexpression screen were obtained from the Broad Institute. Dr. Jeffrey Martell (MIT) provided the pDisplay-mCherry-KDEL plasmid.

### Summary of plasmids

| Name | Features | Promoter/Vector | Details |
|---|---|---|---|
| ERM-APEX2 | C1$_{(1-27)}$-APEX2-V5-Stop | CMV/pLX304 | Soybean APEX2 has four mutations relative to wild-type ascorbate peroxidase: K14D, W41F, E112K, and A134P (*Lam et al., 2015*). C1$_{(1-27)}$ indicates the first 27 amino acids from rabbit cytochrome P450 2C1 (*Ahn et al., 1993*): MDPVVVLGLCLSCLLLLSLW KQSYGGG V5: GKPIPNPLLGLDST This lentiviral vector carries a blasticidin resistance marker. |
| APEX2-OMM | Flag-APEX2-MAVS$_{(510-540)}$-Stop | CMV/pLX304 | Flag: DYKDDDDK MAVS$_{(510-540)}$ (*Seth et al., 2005*): RPSPGALWLQVAVTGVLVVT LLVVLYRRRLH This lentiviral vector carries a blasticidin resistance marker. |
| APEX2-NES | Flag-APEX2-NES-Stop | CMV/pLX304 | NES: LQLPPLERLTLD This lentiviral vector carries a blasticidin resistance marker. |
| SYNJ2BP-V5 | SYNJ2BP-linker-V5-Stop | CMV/pLX_TRC208 | The SYNJ2BP gene was originally obtained as SYNJ2BP-linker-V5 in pLX304 from the Broad Institute. We made a CRISPR-resistant version and moved the construct to this pLX_TRC208. Linker: CPTFLYKVV This lentiviral vector carries a hygromycin resistance marker. |
| mito-tagBFP | mito-tagBFP-Stop | CMV/pcDNA3 | The tagBFP (blue fluorescent protein) gene was cloned from pHR-SFFV-dCas9-BFP, a gift from Stanley Qi and Jonathan Weissman (Addgene plasmid # 46910) (*Gilbert et al., 2013*). Mitochondrial matrix targeting sequence ('mito'): MLATRVFSLVGKRAISTSVCVR |
| V5-GFP-RRBP1 | Calreticulin signal peptide-V5-GFP-RRBP1-Stop | CMV/pcDNA4 HisMax | GFP (green fluorescent protein)-RRBP1 in pcDNA4 HisMax was a gift from Alexander F. Palazzo (University of Toronto) (*Cui et al., 2012*). The RRBP1 gene is the full-length version with 54 decapeptide repeats and the three amino acids after the decapeptide repeat region. |

*Continued on next page*

*Continued*

| Name | Features | Promoter/Vector | Details |
|------|----------|-----------------|---------|
| V5-APEX2-NLS | V5-APEX2-NLS-Stop | CMV/pcDNA3 | NLS: SRADPKKKRKVDPKKKRKVD PKKKRKV |
| Flag-APEX2-NLS | Flag-APEX2-NLS-Stop | CMV/pcDNA3 | |
| Flag-APEX2-CAAX | Flag-APEX2-CAAX-Stop | CMV/pcDNA3 | CAAX: RSKLNPPDESGPGCMSCKCV LS |

## Mammalian cell culture

Human embryonic kidney (HEK) 293T, HeLa, or COS-7 cells were obtained from ATCC. These cells were not independently authenticated or tested for mycoplasma. All lines were cultured in a 1:1 Dulbecco's Modified Eagle's Medium (DMEM) with 4.5 g/l glucose and L-glutamine:Minimum Essential Medium (MEM) mixture supplemented with 10% fetal bovine serum (FBS), 100 units/ml penicillin, and 100 μg/ml streptomycin at 37°C under 5% $CO_2$. For confocal microscopy experiments, cells were plated on 7 mm x 7 mm glass coverslips pre-coated for at least 20 min at 37°C under 5% $CO_2$ with 50 μg/ml human fibronectin (Millipore) in Dulbecco's Phosphate Buffered Saline (DPBS).

## Fluorescence microscopy

Confocal microscopy was performed using a Zeiss AxioObserver.Z1 microscope equipped with a Yokogawa spinning disk confocal head, Cascade IIL:512 camera, and a Quad-band notch dichroic mirror (405/488/568/647 nm). The samples were excited using solid state 405 nm, 491 nm, 561 nm, and 640 nm lasers. Slidebook 5.0 software (Intelligent Imaging Innovations) was used to collect the images through a 48x or 63x oil-immersion objective for BFP (445/40 emission filter), YFP/AF488 (528/38 emission filter), AF568 (617/73 emission filter), AF647 (700/75 emission filter), and differential interference contrast (DIC) channels. Acquisition times ranged from 100 to 2000 ms.

## Characterization of ERM-APEX2 and APEX2-OMM localization by fluorescence microscopy (related to *Figure 1B*)

COS-7 cells were plated on human fibronectin-coated glass coverslips in wells of a 48-well plate so they were at 30–40% confluent the next day. Then, 30 μl of the ERM-APEX2 or APEX2-OMM lentivirus that was used to generate the stable cells for proteomics was combined with 170 μl of full cell culture media each and added to the cells. The cells were transduced for 48 hr, after which they were washed three times with warm DPBS and fixed with 4% paraformaldehyde in 'fixation buffer' (60 mM PIPES, 25 mM HEPES, 10 mM EGTA, 2 mM $MgCl_2$, 120 mM sucrose, pH 7.4) for 15 min at room temperature. The samples were washed three times with 1x phosphate buffered saline (PBS) and then permeabilized by adding cold methanol and placing the plate at −20°C for 5 min. The samples were washed three times with 1x PBS and then blocked for 1 hr at room temperature in 3% bovine serum albumin (BSA) in 1x PBS.

Samples transduced with ERM-APEX2 lentivirus were rocked for 1 hr at room temperature in 1:1000 mouse anti-V5 antibody (Life Technologies, catalog no. R960-25) and 1:50 rabbit anti-RCN2 antibody (Proteintech, catalog no. 10193–2-AP) in 1% BSA in 1x PBS. Samples transduced with APEX2-OMM lentivirus were rocked for 1 hr at room temperature in 1:500 mouse anti-Flag (Agilent, catalog no. 200472) and 1:400 rabbit anti-Tom20 (Santa Cruz Biotechnology, catalog no. sc-11415) in 1% BSA in 1x PBS. The samples were then washed four times with 1x PBS for five minutes each time. All wells were then rocked in 1:2000 anti-mouse Alexa Fluor 568 (AF568), 1:1000 anti-rabbit AF488 in 1% BSA in 1x PBS for 30 min at room temperature. The cells were washed again four times with 1x PBS for five minutes each time. The last wash was replaced with fresh 1x PBS, and the cells were imaged using confocal microscopy. The data in this figure are representative of three independent experiments with ≥10 fields of view each.

## Characterization of ERM-APEX2 and APEX2-OMM by electron microscopy (related to *Figure 1C*)

ERM-APEX2 and APEX2-OMM lentiviruses were prepared as previously described (*Lam et al., 2015*). Briefly, HEK 293T cells were plated at 60–70% confluency in 6-well plates and transfected with 1000 ng of either pLX304-ERM-APEX2 or pLX304-APEX2-OMM, 900 ng dr8.91, and 100 ng of pVSV-G with 10 µl of Lipofectamine 2000 (Invitrogen) in MEM. The transfection solution was replaced with full cell culture media after 3 hr. About 60 hr post-transfection, the cell culture supernatant was filtered through a 0.45 µm filter and used to transduce fresh HEK 293T at 50% confluency in poly-D-lysine-coated glass-bottom dishes that were pretreated with 50 µg/ml human fibronectin. In parallel, a separate sample of HEK 293T cells were plated at 60–80% confluence and transfected with 0.7 µl Lipofectamine 2000 and 100 ng of pcDNA3-APEX2-CAAX (*Lam et al., 2015*). The untransfected cells in this sample served as a negative control.

After 48 hr of transduction with lentivirus or 18–24 hr post-transfection, the cells were processed according to previously published protocols (*Lam et al., 2015*; *Martell et al., 2012*). In brief, room temperature 2% glutaraldehyde (Electron Microscopy Sciences) in sodium cacodylate buffer (100 mM sodium cacodylate with 2 mM $CaCl_2$, pH 7.4) was added to the samples and then replaced with fresh 2% glutaraldehyde in sodium cacodylate buffer. The samples were then moved to ice and kept on ice until resin infiltration. After one hour, the cells were rinsed with two-minute incubations of chilled sodium cacodylate buffer five times. Unreacted glutaraldehyde was quenched by treating the samples with 30 mM glycine in chilled sodium cacodylate buffer for 5 min. The cells were rinsed again with two-minute washes of chilled sodium cacodylate buffer five times. A solution of freshly-diluted 1.4 mM 3,3'-diaminobenzidine (DAB (Sigma), from a stock of free base dissolved in 0.1 M HCl) and 10 mM $H_2O_2$ in chilled sodium cacodylate buffer was added to the cells for 5 min. The DAB solution was removed, and the samples were rinsed five times with two-minute washes of chilled sodium cacodylate buffer and then stained with 2% osmium tetroxide (Electron Microscopy Sciences) in chilled sodium cacodylate buffer for 30 min. The cells were rinsed five times with chilled Millipore water for two minutes each time. Chilled 2% uranyl acetate (Electron Microscopy Sciences) was added to the cells, and the sample was incubated at 4°C overnight. Then, the samples were dehydrated in a cold graded ethanol series (20%, 50%, 70%, 90%, 100%, 100%) for 2 min each. The samples were finally rinsed once in room temperature anhydrous ethanol. They were then infiltrated in Durcupan ACM resin (Electron Microscopy Sciences) in 1:1 anhydrous ethanol:resin for 30 min, then in 100% resin twice for 1 hr each time, before finally exchanging with fresh resin. The samples were then polymerized in a vacuum oven for 48 hr at 60°C. Samples were further processed and imaged as previously described on a JEOL 1200 TEM operating at 80 keV (*Martell et al., 2012*).

The data presented in this figure are representative of ≥3 fields of view from one experiment for each sample.

## Generation and concentration of ERM-APEX2, APEX2-OMM, and APEX2-NES lentiviruses for proteomic samples and fluorescence microscopy (related to *Figure 1B, D and E*, and *Figure 1—figure supplement 1*)

HEK 293T cells were plated in T150 flasks to make lentiviruses for each of the three APEX2 constructs. The cells in each T150 flask were transfected with 120 µl of polyethylenimine (Sigma), 15 µg of pLX304 plasmid carrying an APEX2 construct, 13.5 µg of dr8.91, and 1.5 µg of pVSV-G in MEM. MEM was added to a final volume of 15 ml. This mixture was added to cells for 6 hr at 37°C, 5% $CO_2$, after which it was removed and replaced with 22 ml of full cell culture media. After approximately 40 hr at 37°C, 5% $CO_2$, the supernatant containing lentivirus was filtered using a 0.45 µm filter and stored at 4°C until needed.

The filtered supernatant was then divided into 10 ml fractions and concentrated by ultracentrifugation at 25,000xg for 2 hr at 4°C. The supernatant was decanted, and the tube was dried by inverting the tube and blotting on an ethanol-soaked paper towel for ~1 min. Each fraction of lentivirus was resuspended in 500 µl DPBS by careful pipetting to avoid bubbles, and all of the fractions were pooled together and separated into 120 µl aliquots in 0.6 ml low binding tubes (Fisher Scientific, catalog no. 07-200-183). The tubes were frozen in liquid nitrogen and stored at −80°C.

## Generation of ERM-APEX2, APEX2-OMM, and APEX2-NES stable cell lines (related to *Figure 1D and E*, and *Figure 1—figure supplement 1*)

Low passage HEK 293T cells were plated at ~30% confluency in 6-well plates. Each well was transduced with 30 µl of concentrated lentivirus for ERM-APEX2, APEX2-OMM, or APEX2-NES. After 47 hr at 37°C, 5% $CO_2$, each well was split into two T25 flasks in media containing 8 µg/ml blasticidin S HCl. Each day, the media was changed to fresh 8 µg/ml blasticidin S HCl media. Cells were split and expanded before reaching full confluency. After 8–11 days, one T75 flask for each cell line was made into cell stocks, after which the cells were passaged in drug-free media.

## Characterization of ERM-APEX, APEX2-OMM, and APEX2-NES biotinylation by fluorescence microscopy (related to *Figure 1—figure supplement 1*)

HEK 293T cells stably expressing ERM-APEX2, APEX2-OMM, APEX2-NES, or no APEX2 construct were grown in SILAC media for 11 days (see 'SILAC labeling of proteomic samples, biotinylation, and processing for mass spectrometry' below) and plated on human fibronectin-coated glass coverslips in 48-well plates. The next day, the cells were incubated with 200 µl of 500 µM biotin-phenol in media for 30 min at 37°C, 5% $CO_2$. Hydrogen peroxide ($H_2O_2$) was then added to a final concentration of 1 mM for 1 min at room temperature to initiate the biotinylation reaction. The reaction was then quenched by washing three times with 'quencher solution' (10 mM sodium ascorbate, 10 mM sodium azide, and 5 mM Trolox in DPBS). The cells were then fixed with 3.7% formaldehyde in DPBS for 15 min at room temperature. After fixation, the cells were washed three times with DPBS. Cold methanol was added to permeabilize the cells, and the samples were placed at −20°C for five minutes. The samples were then washed three times with DPBS. The cells were blocked for 1 hr at 4°C with 3% BSA in DPBS.

The samples were rocked in primary antibody (1:500 mouse anti-Flag for APEX2-OMM and APEX2-NES, 1:1000 mouse anti-V5 for ERM-APEX) in 3% BSA in DPBS for 1 hr at 4°C. After primary antibody incubation, the samples were washed three times with 0.2% Tween-20 in DPBS. The samples were then rocked in secondary antibody (1:750 anti-mouse AF488) and 1:1000 neutrAvidin-AF647 in 3% BSA in DPBS for 1 hr at 4°C. The neutrAvidin-AF647 conjugate was prepared following Invitrogen's instructions. Finally, the samples were washed three times with 0.2% Tween-20 in DPBS, maintained in DPBS, and imaged using confocal microscopy.

The data in this figure are representative of three independent experiments with ≥10 fields of view each.

## Western blot analysis of biotin-phenol labeling (related to *Figure 1D*)

Wild-type HEK 293T cells or HEK 293T cells stably expressing ERM-APEX2, APEX2-OMM, or APEX2-NES were plated in T25 flasks. The next day, the cells were pre-incubated with 4 ml 500 µM biotin-phenol in cell culture media for 30 min at 37°C, 5% $CO_2$. Then, $H_2O_2$ was added to a final concentration of 1 mM for 1 min at room temperature. Negative control samples that omitted biotin-phenol or $H_2O_2$ were included for each APEX2-expressing cell line. The reaction was quenched by washing three times with 5 ml of quencher solution. The cells were collected in 5 ml of quencher solution by gently pipetting and then pelleted by centrifuging at 3,000xg for 10 min at 4°C. The supernatant was removed, and the cells were stored at −80°C.

The pellets were thawed on ice and then lysed by gently pipetting in 250 µl of RIPA lysis buffer (50 mM Tris, pH 7.5, 150 mM NaCl, 0.1% SDS, 0.5% sodium deoxycholate, and 1% Triton X-100) with 1x protease inhibitor cocktail (Sigma Aldrich, catalog no. P8849), 1 mM PMSF (phenylmethylsulfonyl fluoride), 10 mM sodium azide, 10 mM sodium ascorbate, and 5 mM Trolox. The lysates were incubated on ice for approximately 1 min and then clarified by centrifuging at 15,000xg for 10 min at 4°C. The clarified whole cell lysates were moved to fresh tubes. The lysates were combined with protein loading buffer, boiled for 10 min, and then separated on a 9% sodium dodecyl sulfate (SDS) gel.

Transfer to a nitrocellulose membrane, Ponceau staining, and blotting with streptavidin-HRP were performed as previously described (*Hung et al., 2014*).

The data in this experiment are representative of two independent experiments.

## Silver stain analysis of streptavidin enriched proteins (related to *Figure 1E*)

Biotinylated whole cell lysate samples were prepared as described in 'Western blot analysis of biotin-phenol labeling,' except the cells were grown in 6-well plates, and the cell pellets were lysed in 100 µl of RIPA lysis buffer with 1x protease inhibitor cocktail, 1 mM PMSF, 10 mM sodium azide, 10 mM sodium ascorbate, and 5 mM Trolox. About 10 µl of each whole cell lysate was set aside to do a western blot to check for biotinylation. A magnetic rack was used to remove the buffer from 550 µl of streptavidin-coated magnetic bead slurry (Pierce, catalog no. 88817). The beads were washed twice and resuspended with 550 µl RIPA lysis buffer. Pipette tips used to handle the beads had their tips sawed off with a clean razor. The remaining 90 µl of whole cell lysate for each sample was incubated with 50 µl of the streptavidin-coated magnetic beads. To facilitate rotation, an additional 500 µl of RIPA was added to each tube. The tubes were rotated at room temperature for 1 hr at room temperature to enrich for biotinylated proteins.

A magnetic rack was used to pellet the beads and collect the supernatant ('flowthrough'). The beads were washed twice with RIPA lysis buffer, once with 1 M potassium chloride (KCl) in Millipore water, once with 0.1 M sodium carbonate ($Na_2CO_3$) in Millipore water, once with 2 M urea in 10 mM Tris-HCl, pH 8.0, and twice more with RIPA lysis buffer using 1 ml of buffer per wash. These wash buffers were kept on ice throughout the procedure.

To elute the enriched proteins, the beads were boiled in 30 µl of 3x SDS protein loading buffer with 2 mM biotin and 20 mM dithiothreitol (DTT) for 10 min at 95°C. The samples were briefly vortexed, cooled on ice, and briefly spun down. The beads were then isolated using a magnetic rack, and the supernatant, 'streptavidin enriched eluate,' was collected and separated on 9% SDS gels. The gels were then analyzed by silver stain (Pierce, cat. no. 24612).

This silver stain was performed once as described above and once with the samples generated for the proteomic experiment.

## SILAC labeling of proteomic samples, biotinylation, and processing for mass spectrometry

Young passage (passage number <12) APEX2-OMM, ERM-APEX2, and APEX2-NES stable HEK 293T cells, and wild-type HEK 293T cells, were seeded into T25 flasks. The cells were then split and cultured in SILAC media (*Ong et al., 2002*), which consisted of DMEM lacking lysine and arginine (Caisson Laboratories) with 10% dialyzed fetal bovine serum (Sigma), 100 units/ml penicillin, 100 µg/ml streptomycin, glutamine, and 4.5 g/l glucose supplemented with different isotopes of L-arginine and L-lysine. Heavy SILAC media contained L-arginine [$^{13}C_6$,$^{15}N_4$]HCl (Arg-10) and L-lysine [$^{13}C_6$,$^{15}N_2$]HCl (Lys-8) (Sigma); medium SILAC media contained L-arginine [$^{13}C_6$]HCl (Arg-6) and L-lysine-4,4,5,5-$d_4$ (Lys-4) (Sigma); light SILAC media contained light L-arginine (Arg-0) and L-lysine (Lys-0) (Sigma). Cell cultures were established as illustrated in *Figure 2A* and as previously described (*Hung et al., 2016*, *Hung et al., 2014*): experimental APEX2-OMM and ERM-APEX2 sample cells were grown in heavy SILAC media, APEX2-NES cells were grown in medium SILAC media, and negative control cells (where either APEX2 expression or $H_2O_2$ was omitted) were grown in light SILAC media. Spike-in samples, where APEX2-OMM and ERM-APEX2 stable cells were treated identically to their heavy experimental sample counterparts, were grown in light SILAC media. Cells were passaged every 2–3 days before reaching confluency such that after 11 days, there was one T75 flask of each spike-in sample grown in light SILAC media, two T150 flasks of APEX2-OMM cells grown in heavy SILAC media, two T150 flasks of ERM-APEX2 cells grown in heavy SILAC media, and four T150 flasks of APEX2-NES cells grown in medium SILAC media.

To biotinylate the proteomic samples, the media in each T150 flask was changed to 30 ml of 500 µM biotin-phenol in its respective SILAC media, and the flasks were returned to the incubator (37°C, 5% $CO_2$) for 30 min. All subsequent steps were performed at room temperature. First, the flasks were removed from the incubator and turned upside down so the media pooled on the ceiling of the flasks. $H_2O_2$ was added to the media to a final concentration of 1 mM, and the flasks were briefly agitated to mix. The flasks were then inverted to expose the cells to the media with biotin-phenol and $H_2O_2$ for 1 min. The flasks were then turned upside down again, and the media was poured out. Using the same flask inversion technique to facilitate rapid solution transfer without dislodging cells, the cells were washed for 1 min in 25 mL of quencher solution, followed by rapid 25 mL washes as

follows: once with quencher solution, twice with DPBS, and once more with quencher solution. The cells were then resuspended in 7 mL fresh quencher solution by pipetting, and the cells were pelleted by centrifuging at 10,000xg for 10 min at 4°C. The supernatant was removed, and the cell pellets were stored at −80°C overnight. The spike-in samples were labeled identically using half of each volume that was used for a T150 flask. The cells from the spike-in samples were resuspended, pelleted, and stored identically.

The next day, the pellets were thawed on ice and lysed in 800 µl fresh RIPA buffer with 10 mM sodium ascorbate, 10 mM sodium azide, 5 mM Trolox, 1x protease inhibitor cocktail, and 1 mM PMSF by gentle pipetting. The pellets for the spike-in samples were lysed in half the volume. The lysates were clarified by centrifuging at 15,000xg for 10 min at 4°C. The clarified lysates were then moved to fresh tubes.

A Pierce 660 nm kit (Thermo Fisher Scientific) was used to measure the protein concentration of each sample, and the lysates from the heavy, medium, and light samples were combined 1:1:1 by protein mass as indicated in *Figure 2A*. The amount of protein used in each experiment was determined by whichever sample was limiting, with the maximum amount of 1:1:1 sample available used for each replicate. For Replicate 1 of the OMM proteomic experiment, 1.29 mg of protein for each SILAC state was used; for Replicate 2 of the OMM proteomic experiment, 2.5 mg of protein for each SILAC state was used; for both replicates of the ERM proteomic experiment, 2.7 mg of protein for each SILAC state was used. Spike-in samples were added to 5% of the protein mass used for the heavy state in each of the two replicates for the OMM and ERM proteomic experiments, respectively.

Each of the four resulting whole cell lysate mixtures was incubated with 500 µl of streptavidin-coated magnetic beads that were washed twice with RIPA lysis buffer. The samples were rotated at room temperature for 1 hr and washed as described in 'Silver stain analysis of streptavidin enriched proteins' with 1 mL washes per mixture. The biotinylated proteins were eluted from the beads by boiling for 10 min at 95°C in 65 µl of 3x SDS protein loading buffer supplemented with 2 mM biotin and 20 mM DTT. The samples were vortexed, cooled on ice, and briefly spun down. A magnetic rack was used to pellet the beads. The supernatant was collected and moved to a fresh tube.

## In gel protein digestion and mass spectrometry of the OMM and ERM proteomic samples

Biotinylated proteins eluted from streptavidin beads were separated on a NuPAGE Novex Bis-Tris 4–12% gel (Thermo Fisher Scientific) for 1 hr at 130V. The gel was stained overnight with SimplyBlue Coomassie SafeStain (Thermo Fisher Scientific) and washed with water for several hours. For each gel lane, 16 gel bands were excised, cut into ~1 mm² bits, and transferred to fresh tubes. Gel pieces were washed once with 200 µl of 100 mM ammonium bicarbonate, pH 8.0. Gel pieces were further destained with 200 µl of 1:1 acetonitrile:100 mM ammonium bicarbonate, pH 8.0 for several hours, washed once with 100 µl of 100 mM ammonium bicarbonate, pH 8.0, and dehydrated for 5 min with 100 µl of 100% acetonitrile. Dehydrated gel pieces were incubated with 100 µl of 10 mM DTT in 100 mM ammonium bicarbonate, pH 8.0 for one hour with shaking, followed by 100 µl of 55 mM iodoacetamide for 45 min in the dark, and then dehydrated with 100 µl of 100% acetonitrile. Proteins were digested overnight by adding 10–50 µl of 10 ng/µl sequencing grade Trypsin (Promega) to the gel pieces. Peptides were extracted by incubating the gel pieces three times with 15–20 µl of 60% MeCN:0.1% trifluoracetic acid (TFA) solution and then one time with 15–20 µl of 100% acetonitrile. The extraction solution was dried down by vacuum centrifugation. Samples were desalted exactly as previously described using C18 stage tips (*Hung et al., 2016*).

Desalted peptides were resuspended in 9 µl of 3% MeCN/0.1% formic acid and analyzed by online nanoflow liquid chromatography tandem mass spectrometry (LC-MS/MS) using a Q Exactive mass spectrometer (Thermo Fisher Scientific) coupled online to a Proxeon Easy-nLC 1000 (Thermo Fisher Scientific) as previously described (*Hung et al., 2016*). Briefly, 4 µl of each sample was loaded on to a microcapillary column (360 µm outer diameter x 75 µm inner diameter) containing an integrated electrospray emitter tip (10 µm) and packed to approximately 22 cm with ReproSil-Pur C18-AQ 1.9 µm beads (Dr. Maisch GmbH). The column was heated to 50°C. The HPLC solvent A was 0.1% formic acid, and solvent B was 90% MeCN/0.1% formic acid. Peptides were eluted into the mass spectrometer at a flow rate of 200 nl/min. A 120 min inject-to-inject LC-MS method with a linear gradient of 0.3% B/min for 82 min, followed by a ramp to 60% B (10% B/min) was used. The Q

Exactive was operated in the data-dependent mode acquiring HCD MS/MS scans (R = 17,500) after each MS1 scan (R = 70,000) on the top 12 most abundant ions using an MS1 target of $3 \times 10^6$ and an MS2 target of $5 \times 10^4$. The maximum ion time utilized for MS/MS scans was 120 ms; the HCD-normalized collision energy was set to 25; the dynamic exclusion time was set to 20 s; and the peptide match and isotope exclusion functions were enabled.

MS data were analyzed using MaxQuant software version 1.3.0.5 (*Cox and Mann, 2008*). Files were searched against the human Uniprot database, which also included a list of common laboratory contaminants provided by MaxQuant. The search parameters were set as follows. The enzyme specificity was set to trypsin with the maximum number of missed cleavages set to 2. The precursor mass tolerance was set to 6 ppm. Variable modifications searched were oxidized methionine, N-terminal protein acetylation, and biotin-phenol on Y. Carbamidomethylation of cysteine was searched as a fixed modification. The protein false discovery rate was set to 1%. Protein identification and quantification information were obtained from the MaxQuant 'protein groups' table.

The proteomic experiment was performed once.

## Determination of SILAC ratio cut-offs (related to *Figure 2—figure supplement 1*)

Each of the two replicates for the OMM and ERM proteomic experiments was analyzed separately. Proteins marked as contaminants or reverse hits by MaxQuant software (*Cox et al., 2011*; *Cox and Mann, 2008*) were removed. 'Detected proteins,' which we defined as those with at least two unique, sequenced peptides, were retained for further analysis. To normalize the SILAC ratios, we used the unnormalized protein SILAC ratios from Maxquant and calculated the median of the distribution of heavy/light (H/L) or heavy/medium (H/M) ratios for the subset of soluble mitochondrial matrix proteins (*Rhee et al., 2013*) detected in each experiment. We then divided the ratio for each detected protein by this median and then took the $\log_2$ of the normalized ratios. The $\log_2$(normalized ratios) were used for all other analysis, and only proteins possessing both H/L and H/M ratios in both replicates were used for subsequent analysis.

To obtain $\log_2$(H/L) cut-offs for the OMM proteomic experiment, proteins were classified as (1) those in our curated list of known OMM proteins (*Supplementary file 2a*), (2) soluble mitochondrial matrix proteins (*Rhee et al., 2013*), or (3) all others. Class (1) proteins are true positive proteins, and class (2) proteins are false positive proteins. The proteins were ranked from high to low $\log_2$(H/L) ratio. Then, for every possible SILAC ratio cut-off, we calculated the true positive rate (TPR) and the false positive rate (FPR). We defined TPR as the fraction of detected true positives above the cut-off and FPR as the fraction of detected false positives above the cut-off. The $\log_2$(H/L) ratio cut-offs were selected where the difference between the TPR and FPR was maximal. The $\log_2$(H/L) ratio cut-offs for the ERM proteome were calculated in the same way, except the true positives were known ERM proteins (*Supplementary file 2b*).

To obtain $\log_2$(H/M) cut-offs for the OMM proteomic experiment, the proteins were classified as (1) proteins belonging to our curated list of known OMM proteins (*Supplementary file 2a*), (2) proteins that lack prior mitochondrial annotation according to MitoCarta (*Pagliarini et al., 2008*), GOCC (*Ashburner et al., 2000*), our previous mitochondrial matrix (*Rhee et al., 2013*) or mitochondrial intermembrane space (IMS) (*Hung et al., 2014*) proteomes, or literature, or (3) all others. The proteins were ranked from high to low $\log_2$(H/M) ratio. For this analysis, the false positives were the class two proteins. TPR and FPR were calculated as above, and the cut-offs were also selected by the same criteria. A similar analysis was used for the $\log_2$(H/M) cut-offs for the ERM proteome, except the true positives were known ERM proteins (*Supplementary file 2b*), and the false positives were all proteins that were not on our curated list of known ERM proteins.

Our proteomes consisted of proteins that fell above the $\log_2$(H/L) and $\log_2$(H/M) cut-offs in both replicates of the proteomic experiment. These cut-offs (truncated to two decimal points) are shown in the table below.

## SILAC ratio cut-offs used to define the final proteomes

| Proteome | Replicate | $\log_2$(H/L) cut-off | $\log_2$(H/M) cut-off |
|---|---|---|---|
| OMM | 1 | 1.06 | 0.68 |
|  | 2 | 1.07 | 0.74 |
| ERM | 1 | 0.49 | 0.08 |
|  | 2 | 0.37 | 0.10 |

## Scatter plot analysis (related to *Figure 2B*)

All proteins in Replicate 1 of the OMM proteomic experiment were classified as: (1) known OMM proteins (*Supplementary file 2a*), (2) proteins that lack prior mitochondrial annotation by MitoCarta (*Pagliarini et al., 2008*), GOCC (*Ashburner et al., 2000*), our previous mitochondrial matrix (*Rhee et al., 2013*) or mitochondrial IMS (*Hung et al., 2014*) proteomes, or literature, or (3) all others. In the scatter plot of $\log_2$(H/M) versus $\log_2$(H/L), class (1) proteins are plotted in green, class (2) proteins are plotted in red, and class (3) proteins are plotted in black.

Similarly, proteins in Replicate 1 of the ERM proteomic experiment were separated into three classes: (1) known ERM proteins (*Supplementary file 2b*), (2) proteins with the gene ontology (GO) term GO:0005829 for cytosol that lacked annotated or predicted transmembrane domains according to UniProt or the transmembrane hidden Markov model (TMHMM) (*Krogh et al., 2001*; *Sonnhammer et al., 1998*), and (3) all other proteins. As before, class (1) proteins are colored in green, class (2) proteins are colored in red, and class (3) proteins are colored in black.

## Specificity analysis (related to *Figure 2C*)

We retrieved the human proteome from the UniProt-GO Annotation database in April 2014. To generate column 1, we generated a list of mitochondrial GO terms from the AmiGo database by searching for 'mitochon' and only retaining GOCC terms. Cell-type specific terms such as those that were specific to sperm were discarded. Proteins that had at least one of these mitochondrial GOCC terms or were present in MitoCarta (*Pagliarini et al., 2008*), our previous mitochondrial matrix proteome (*Rhee et al., 2013*), or our previous IMS proteome (*Hung et al., 2014*) were labeled as mitochondrial. In addition, we also included proteins in the OMM proteome that were identified as mitochondrial in literature. For column 2, we asked which proteins in the OMM proteome had prior mitochondrial annotation according to the same metrics.

Sub-mitochondrial specificity of the OMM proteome (column 4) was calculated by determining which proteins had the following GO terms: GO:0005741 for OMM, GO:0005758 for IMS, GO:0005743 for IMM, and GO:0005759 for mitochondrial matrix. Any protein with more than one sub-mitochondrial annotation was assigned to one compartment according to this priority: OMM>IMS>IMM>mitochondrial matrix; that is, if a protein had both OMM and IMS annotation, it would be classified as an OMM protein. For comparison, we show the same analysis for all known mitochondrial proteins that have these annotations available in column 3.

For secretory pathway specificity, we generated a list of GO terms by searching for the following terms in the AmiGO database and retaining terms that included the words used to find them: 'endoplasmic reticulum,' 'ER,' 'golgi,' 'plasma membrane,' 'extracellular,' 'endosom,' 'lysosom,' 'nuclear envelope,' 'nuclear membrane,' 'perinuclear region of cytoplasm,' and 'vesicle.' In addition, we used the list of proteins predicted to have transmembrane domains or signal peptides according to Phobius (*Käll et al., 2007*, *Käll et al., 2004*). The percentage of proteins in the human proteome (column 5) and our ERM proteome (column 6) with these GO terms or with Phobius annotation was calculated.

For sub-secretory pathway specificity of the human proteome (column 7), we took the subset of proteins with the following GO terms: GO:0005783 for endoplasmic reticulum, GO:0005794 for Golgi apparatus, and GO:0005886 for plasma membrane and classified them according to this priority: endoplasmic reticulum>Golgi apparatus>plasma membrane. We then took the subset of proteins in the ERM proteome with these GO terms and plotted their percentages in column 8.

## Characterization of endogenous C2CD3 expression by fluorescence microscopy (related to *Figure 2D*)

COS-7 cells were plated at ~70% confluency on 50 µg/ml human fibronectin-coated glass coverslips in a 48-well plate. The next day, the cells were transfected with 100 ng of mito-tagBFP, 100 ng of mCherry-KDEL, and 0.8 µl of Lipofectamine 2000 in MEM for four hours. The transfection solution was then replaced with full cell culture media. About 24 hr post-media change, the cells were washed three times with warmed DPBS. The cells were fixed with 250 µl of 4% paraformaldehyde in fixation buffer for 15 min at room temperature and then washed three times with 1x PBS. The sample was permeabilized with cold methanol, and the plate was placed at −20°C for 5 min. Then, the cells were washed three times with 1x PBS and blocked overnight at 4°C in 3% BSA in 1x PBS.

The next day, the cells were rocked in 1:250 rabbit anti-C2CD3 (Novus Biologicals NBP1-94074) in 1% BSA in 1x PBS for one hour at room temperature. The sample was then washed with 1x PBS four times for five minutes each time. Then, the cells were rocked in 1:1000 anti-rabbit AF488 in 1% BSA in 1x PBS for 30 min at room temperature. Finally, the cells were washed again four times for five minutes each time in 1x PBS. The final wash was removed and replaced with fresh 1x PBS, and the sample was imaged by confocal microscopy.

The image shown in this figure is representative of 22 fields of view from one experiment.

## Characterization of SYNJ2BP-V5 overexpression by fluorescence microscopy (related to *Figure 3A*)

COS-7 cells were plated at ~70% confluency on glass cover slips pre-coated with 50 µg/ml human fibronectin. The next day, the cells were transfected with 200 ng of pLX304-SYNJ2BP-V5 and 150 ng of mCherry-KDEL with 0.8 µl Lipofectamine 2000 in MEM for 4 hr before media change to full cell culture media. About 26 hr later, the cells were washed three times with DPBS and fixed with 4% paraformaldehyde in fixation buffer for 15 min at room temperature. The sample was then washed three times with 1x PBS and permeabilized with cold methanol. The plate was placed at −20°C for 5 min. The samples were then washed three times with 1x PBS and blocked overnight in 3% BSA in 1x PBS at 4°C.

The cells were then incubated in 1:1000 mouse anti-V5 antibody and 1:400 rabbit anti-Tom20 in 1% BSA in 1x PBS for 1 hr at room temperature with gentle rocking. The samples were washed four times with 1x PBS for 5 min each time. Then, the cells were rocked in 1:2000 anti-mouse AF488 and 1:1000 anti-rabbit AF647 in 1% BSA in 1x PBS for 30 min at room temperature. The samples were washed 4 times with 1x PBS for 5 min each time. The last wash was removed and replaced with fresh 1x PBS, and the samples were imaged using confocal microscopy.

These images are representative of five independent experiments with at least 14 fields of view per experiment.

## Characterization of endogenous SYNJ2BP, endogenous RRBP1, and transfected V5-GFP-RRBP1 by fluorescence microscopy (related to *Figure 3—figure supplement 1A* and *Figure 4—figure supplement 1C*)

COS-7 cells were plated on 50 µg/ml human fibronectin-coated glass coverslips in a 48-well plate. The next day, the cells were ~70% confluent. All wells were transfected with 100 ng of mito-tagBFP, 100 ng of mCherry-KDEL, and 0.8 µl of Lipofectamine 2000 in MEM. One well was also co-transfected with 200 ng V5-GFP-RRBP1. After four hours, the transfection solution was removed and replaced with full cell culture media.

About 24 hr later, the cells were washed three times with DPBS. They were then fixed with 250 µl of 4% paraformaldehyde in fixation buffer/well for 15 min at room temperature. The cells were then washed three times with 1x PBS and permeabilized with 200 µl cold methanol/well. The plate was placed at −20°C for 7 min. Then, the samples were washed three times with 1x PBS and blocked overnight in 3% BSA in 1x PBS at 4°C.

The next day, the sample used to detect endogenous SYNJ2BP was rocked in 1:100 rabbit anti-SYNJ2BP antibody (Sigma Aldrich, catalog no. HPA000866) in 1% BSA in 1x PBS for 1 hr at room temperature. The sample used to detect endogenous RRBP1 was rocked in 1:500 rabbit anti-RRBP1 antibody (Abcam, catalog no. 95983) in 1% BSA in 1x PBS for 1 hr at room temperature. Both samples were washed four times for five minutes each time with 1x PBS. Both samples were then rocked

in 1% BSA in 1x PBS containing 1:1000 anti-rabbit AF488 for 30 min at room temperature. All samples were washed four times for five minutes each time with 1x PBS. The final wash was removed and replaced with fresh 1x PBS, and the samples were then imaged using confocal microscopy.

The data in *Figure 3—figure supplement 1A* are representative of two independent experiments with ≥11 fields of view each. The data in *Figure 4—figure supplement 1C* are representative of two independent experiments with ≥17 fields of view each.

## Generation of CRISPR knockout cell lines (related to *Figure 4—figure supplement 1A*)

First, lentivirus was generated using HEK 293T cells as previously described (*Lam et al., 2015*) carrying plentiCas9-Blast (Addgene #52962 (*Sanjana et al., 2014*)). The lentivirus was titered in HeLa and HEK 293T cells to achieve a multiplicity of infection <1. Transduced HeLa and HEK 293T cells stably expressing Cas9 were selected with 8 µg/ml blasticidin S HCl. Three guide RNA sequences from the Broad Institute's Avana library (*Doench et al., 2016*) were selected each for SYNJ2BP and for RRBP1 and cloned into lentiGuide-Puro vectors (Addgene #52963 (*Sanjana et al., 2014*)).

HEK 293T cells were plated in a 24-well plate. The next day, these were transfected with 250 ng of the guide RNA plasmid, 25 ng of pVSV-G, and 225 ng of dr8.91 with 2 µl of Lipofectamine 2000 for 3 hr before changing to full cell culture media. After 1.5 days, fresh HEK 293T or HeLa cells stably expressing Cas9 were plated in 24-well plates at ~30% confluency. The next day, the supernatant of each lentiviral culture was collected and centrifuged at 10,000xg for 10 min. The supernatant, which contained the lentivirus, was transferred to a new tube. Cells stably expressing Cas9 were grown in 250 µl of the lentivirus supplemented with 250 µl of full cell culture media for 2 days, after which they were split and expanded into media containing 8 µg/ml blasticidin S HCl and either 0.5 µg/ml puromycin for HeLa cells or 1 µg/ml puromycin for HEK 293T cells. Media was changed to fresh drug-containing media each day for five days. Knockdown efficiency was assessed by western blotting with an anti-SYNJ2BP antibody (Sigma Aldrich, catalog no. HPA000866) at 1:500 dilution or with an anti-RRBP1 antibody (Abcam, catalog no. 95983) at 1:3000 dilution.

In order to make clonal SYNJ2BP and RRBP1 knockout cell lines, two populations of HEK 293T cells that stably expressed Cas9 and a guide RNA (each population expressed a different guide RNA; see the table below) were counted and plated into duplicate 10-cm cell culture dishes at 100 and 1000 cells per plate in 10 ml media with 1 µg/ml puromycin and 8 µg/ml blasticidin S HCl. The cells were monitored for about two weeks while growing in the incubator, with drug media being added when needed. Individual colonies were first identified under a microscope and encased in cloning cylinders (EMD Millipore, catalog no. TR-1004) before being lifted with trypsin into wells in a 48-well plate and expanded.

## Guide RNA (gRNA) sequences used to generate knockout lines

| Target gene | Guide RNA | Sequence 5'−3' |
| --- | --- | --- |
| None (does not target any sequence in the mammalian genome) | A | ACACCGAAGCACCTGTACGT |
| SYNJ2BP | 1 | AAGAGATCAATCTTACCAGA |
| SYNJ2BP | 2 | GTAAAACATGAACGGAAGAG |
| RRBP1 | 1 | ACAGGAGCAACGATAATGGG |
| RRBP1 | 2 | GGCGTTTCAGAATCGCCACA |

Knockout of SYNJ2BP or RRBP1 was confirmed by western blotting of whole cell lysate. In the case of the blots shown in *Figure 4—figure supplement 1A*, clonal non-targeted Cas9, SYNJ2BP knockout, or RRBP1 knockout HEK 293T cells from wells of a 6-well plate were lysed in 100 µl RIPA buffer with 1x protease inhibitor cocktail and 1 mM PMSF. The lysates were clarified by centrifuging at 15,000xg for 10 min at 4°C, and the supernatant was moved to clean tubes. For each sample, 20 µg of whole cell lysate was combined with SDS protein loading buffer and boiled for 10 min at 95°C. Then, the whole cell lysates were separated either on a 9% SDS gel (to analyze RRBP1 knockout) or a 12% SDS gel (to analyze SYNJ2BP knockout) at 180 V. The proteins were transferred to

nitrocellulose membranes at 400 mA for 2 hr. Ponceau S staining (10 min in 0.1% w/v Ponceau S in 5% acetic acid/water) was used to confirm transfer.

The membrane used to analyze RRBP1 knockout was cut across horizontally. The top half of the membrane was blocked for 1 hr and 12 min in 5% milk in 1x tris-buffered saline with 0.1% Tween-20 (TBST) at room temperature, while the bottom half was blocked in 3% BSA in 1x TBST. The top part was then incubated face down in 1:3000 rabbit anti-RRBP1 (Abcam, catalog no. 95983) for 1 hr at room temperature. The bottom part was rocked in 10 ml of 1:2000 mouse anti-β-actin antibody (Sigma Aldrich, catalog no. A5316) in 3% BSA in 1x TBST. The membranes were washed four times with 1x TBST for five minutes each time. The membranes were then rocked in 10 ml of 3% BSA in 1x TBST with 1:2000 anti-rabbit horseradish peroxidase (HRP) or 10 ml of 1:2000 anti-mouse HRP (Bio-rad), respectively, for 1 hr at room temperature. Then, the membranes were washed four times with 1x TBST for five minutes each time, rinsed with deionized water four times, and developed with Clarity reagent (Bio-rad).

The membrane used to analyze SYNJ2BP knockout was blocked for 1 hr and 12 min in 3% BSA in 1x TBST and then cut across horizontally. The bottom half was incubated face down in 1:500 rabbit anti-SYNJ2BP (Sigma Aldrich, catalog no. HPA000866) in 3% BSA in 1x TBST for 1 hr at room temperature. The top half was incubated in 1:2000 mouse anti-β-actin antibody (Sigma Aldrich, catalog no. A5316) as above. Subsequent washes, secondary antibody incubation, and development with Clarity reagent was performed as above.

The blots were imaged with a UVP gel imager. Knockout of SYNJ2BP or RRBP1 in these cell lines was reconfirmed after the initial characterization either by western blotting or fluorescence imaging using antibodies against endogenous SYNJ2BP or RRBP1, respectively, each time the line was used for an experiment.

## Generation of the stable cell lines for pull down of SYNJ2BP-V5 and immunoprecipitation-mass spectrometry (related to *Figure 3—figure supplement 1C*)

Lentivirus carrying CRISPR-resistant SYNJ2BP-V5 or a control, non-targeted guide RNA was generated by transfecting HEK 293T cells in 6-well plates with 1000 ng of CRISPR-resistant SYNJ2BP-V5 in pLX_TRC208 or the plasmid encoding a guide RNA sequence that does not target the mammalian genome (*Doench et al., 2016*; *Sanjana et al., 2014*), 900 ng of dr8.91, and 100 ng of pVSV-G with 8 µl of Lipofectamine 2000. After four hr of transfection, the media was changed to complete cell culture media. The cells were returned to the incubator for ~2.5 days, after which the supernatant was collected, filtered through a 0.45 µm filter, and stored at 4°C.

To generate HeLa cells stably overexpressing SYNJ2BP-V5, 800 µl of SYNJ2BP-V5 lentivirus and 1200 µl of media was added to one well in a 6-well plate of a population of HeLa cells where SYNJ2BP was knocked down using SYNJ2BP guide RNA-1 and Cas9. After ~2 days of transduction, the cells were split into two T25 flasks in media containing 300 µg/ml hygromycin B to select for transduced cells, and the cultures were grown and expanded. The media was changed to fresh 300 µg/ml hygromycin B media each day for seven days, after which cell stocks were made.

To make HeLa cells expressing a non-targeted Cas9, Cas9 stable cells were plated in a 6-well plate. The next day, the media was replaced with 800 µl of non-targeted guide RNA lentivirus and 1200 µl of media, and the cells were returned to the incubator for two days. The cells were then split into two T25 flasks in media containing 0.5 µg/ml puromycin. The population of cells was then expanded, and the media was changed each day to fresh 0.5 µg/ml puromycin in media for six days, after which cell stocks were made.

To assess the level of SYNJ2BP in these lines, 15 µg of whole cell lysate was combined with SDS protein loading buffer, boiled, cooled, and spun down. The denatured lysates were then separated by gel electrophoresis and transferred to a nitrocellulose membrane. The membrane was blocked overnight at 4°C in 3% BSA in 1x TBST and then incubated face down in 1:500 rabbit anti-SYNJ2BP (Sigma Aldrich, catalog no. HPA000866) in 3% BSA in 1x TBST for 1 hr. Further processing was performed as above. This blot was performed once.

## Pull down of SYNJ2BP-V5 and immunoprecipitation-mass spectrometry (related to *Figure 3C*)

HeLa cells stably expressing non-targeted Cas9, HeLa cells stably overexpressing CRISPR-resistant SYNJ2BP-V5 in a SYNJ2BP knockout background, and a population of Hela cells with SYNJ2BP knocked out were seeded and expanded so there were four T150 flasks for each of the four samples illustrated in *Figure 3—figure supplement 1D*. The cells were then washed twice with 12 ml of DPBS and scraped into 5 ml of DPBS. An additional 5 ml of DPBS was used to wash the flask and collect the remaining cells, which were then added to the suspension of scraped cells. The cells were pelleted at 3,000xg for 10 min at 4°C and stored at −80°C so that there were four pellets for each sample.

The cells were thawed on ice and then lysed in 400 µl co-immunoprecipitation (co-IP) lysis buffer (50 mM Tris-HCl, pH 7.4, 150 mM NaCl, and 0.5% Triton X-100) with 1 mM PMSF and 1x protease inhibitor cocktail. The lysates were clarified by centrifuging at 15,000xg for 10 min at 4°C, and the whole cell lysates for each sample were then combined into one tube. The protein concentrations of these four samples were measured using the bicinchoninic acid (BCA) assay (Thermo Fisher Scientific).

Four aliquots of 100 µl of Dynabeads Protein G (Thermo Fisher Scientific) were each rotated in 400 µl of 1:100 mouse anti-V5 in 1x PBS, pH 7.4 for 30 min at room temperature. A magnetic rack was used to isolate the beads. The beads were then washed twice with 1x PBS, pH 7.4. After the second wash was removed, 1 mg of whole cell lysate was added to the beads for each of the four samples. Co-IP lysis buffer containing 1 mM PMSF and 1x protease inhibitor cocktail was added so that the final volumes for all of the samples were equal. The tubes were rotated at room temperature for 1 hr. The magnetic rack was used to pellet the beads and collect the flowthrough samples, and then the beads were washed three times with co-IP lysis buffer. The samples were transferred to fresh tubes while in the third wash.

Proteins bound to the beads were washed twice with 200 µl of 50 mM Tris-HCl buffer, pH 7.5 and twice with 200 µl of 2 M urea/50 mM Tris buffer, pH 7.5. Proteins were pre-digested for 1 hr at room temperature with 0.4 µg trypsin in 80 µl of 2 M urea/50 mM Tris buffer, pH 7.5 containing 1 mM DTT with shaking at 1000xg. Following pre-digestion, the supernatants were transferred to fresh tubes. Beads were washed twice with 60 µl of 2 M urea/50 mM Tris, and these washes were combined with the supernatant. Eluates were reduced with 1.6 µl of 4 mM DTT for 30 min at room temperature with shaking, followed by alkylation with 4 µl of 10 mM iodoacetamide for 45 min in the dark at room temperature. An additional 0.5 µg of trypsin was added, and samples were digested overnight at room temperature with shaking. Following overnight digestion, samples were acidified (pH <3) with neat formic acid to a final concentration of 1%. Acidified samples were desalted on C18 stage tips as previously described (*Hung et al., 2016*).

Peptides were labeled with 4-plex isobaric Tags for Relative and Absolute Quantification (iTRAQ) reagents in accordance with the manufacturing instructions (Sciex, Foster City, CA). Briefly, peptides were resuspended in 30 µl of dissolution buffer and immediately combined with 70 µl of ethanol. One unit of iTRAQ reagent was used to label each experimental condition in the 4-plex cassette. Samples were labeled as follows: 114-control non-targeted gRNA, 115-SYNJ2BP knockout (via CRISPR/Cas9) cells, 116-SYNJ2BP-V5 stables replicate 1, 117-SYNJ2BP-V5 stables replicate 2. Samples were incubated for 1 hr at room temperature with gentle shaking. Following incubation, approximately 3% of sample was taken to test label incorporation. The remaining iTRAQ labeling reaction was quenched with 10 µl of 1 M Tris-HCl, pH 8.0, for 10 min at room temperature with gentle shaking. Samples were combined, dried in a vacuum centrifuge, and desalted on C18 stage tips as described (*Hung et al., 2016*).

Peptides were resuspended in 9 µl of 3% MeCN/0.1% formic acid, and analyzed by online nanoflow liquid chromatography-tandem mass spectrometry (LC-MS/MS) using a Q Exactive Plus mass spectrometer (Thermo Fisher Scientific) coupled online to a Proxeon Easy-nLC 1000 (Thermo Fisher Scientific) as described above with a few modifications. A 260 min inject-to-inject LC-MS method was utilized. For the 260 min method, after a 1 min ramp to 6% B, a gradient of 0.1% B/minute was applied for 234 min followed by a ramp to 60% B (3.3% B/minute). The HCD-normalized collision energy for the iTRAQ labeled sample was set to 27. iTRAQ data were analyzed using Spectrum Mill software package v5.0 pre-release (Agilent Technologies, Santa Clara, CA), as previously described

(*Loh et al., 2016*). Briefly, nearby MS scans with similar precursor m/z values were merged if they were within ±60 s retention time and ±1.4 m/z tolerance. If MS/MS spectra were not within 750–4000 Da precursor MH+ range or had sequence tag lengths >0, they were excluded from the search parameters. MS/MS spectra were searched against UniProt Human complete isoform database appended with 150 common laboratory contaminants. Carbamidomethylation and iTRAQ labeling of N-terminal and internal lysines were set as fixed modifications. N-terminal acetylation and methionine oxidation were set as variable modifications. Minimum matched peak intensity was set to 30%, with a pre-cursor mass tolerance of ±20 ppm. Individual spectra were automatically assigned a confidence score using the Spectrum Mill autovalidation module. The score at the peptide mode was based on a target-decoy false discovery rate (FDR) of 1%.

For data analysis, contaminants and proteins identified as reverse hits were removed. iTRAQ ratios representing SYNJ2BP-V5 vs non-targeted gRNA (116/114) and SYNJ2BP-V5 vs SYNJ2BP knockout (117/115) were median normalized and compared. For each experiment, only proteins with two or more unique quantified peptides and two or more of both the 116/114 and the 117/115 iTRAQ ratios were considered 'detected' and retained for further analysis. These iTRAQ ratios were subjected to a moderated *t*-test to assess statistical significance (*Smyth, 2004*). This statistic is similar to the ordinary *t*-statistic, with the exception that the standard errors are calculated using an empirical Bayes method using information across all proteins, thereby making inference about each individual protein more robust. The nominal *p*-values arising from the moderated *t*-statistic are corrected for multiple testing by controlling the FDR (*Benjamini and Hochberg, 1995*) Proteins with an FDR-adjusted *p*-value of less than 0.02 were deemed to be reproducibly enriched.

The IP-MS experiment was performed once.

## Western blot analysis of SYNJ2BP-V5 pull downs in the presence of translation inhibitors (related to *Figure 4A*)

HEK 293T cells were plated in a 6-well plate. The next day, three out of the six wells were transfected with 2000 ng of pLX_TRC208-SYNJ2BP-V5 with 8 µl Lipofectamine 2000 in MEM for 4 hr, with the remaining three wells untransfected. After 4 hr, the media was changed back to complete cell culture media. After 24 hr, the media in one transfected well and one untransfected well was replaced with 2 ml of 200 µM cycloheximide in media, and the media in another pair of an untransfected well and a transfected well was replaced with 2 ml of 200 µM puromycin in media. The plate was incubated at 37°C, 5% $CO_2$ for 2 hr. The cells were washed three times with DPBS, collected in 1 ml DPBS, and pelleted by centrifugation at 3000xg for 10 min at 4°C. The supernatant was removed, and the pellets were stored at −80°C overnight.

The pellets were thawed on ice and lysed in 100 µl co-IP lysis buffer with 1 mM PMSF and 1x protease inhibitor cocktail by pipetting. The lysates were incubated on ice for about 1 min and then clarified by centrifuging at 15,000xg for 10 min at 4°C. The clarified whole cell lysates were moved to fresh tubes, and a BCA assay was used to measure their protein concentrations.

Meanwhile, 6 aliquots of 25 µl of Dynabeads Protein G each were rotated in 100 µl of 1:100 mouse anti-V5 in 1x PBS, pH 7.4, at room temperature for 30 min. A magnetic rack was used to pellet the beads and wash them twice with 1x PBS, pH 7.4. Then, 400 µg of whole cell lysate was combined with an aliquot of beads for each sample. Co-IP lysis buffer supplemented with 1 mM PMSF and 1x protease inhibitor cocktail was added to each sample to attain equal volumes. The samples were rotated at room temperature for 1 hr. After 1 hr, the magnetic rack was used to pellet the beads and collect the flowthrough.

The beads were washed three times with co-IP lysis buffer and were transferred to fresh tubes while in the third wash. The wash was removed, and the proteins were eluted by boiling in 15 µl of 3x SDS protein loading buffer for 10 min at 95°C. The tubes were cooled on ice and spun briefly to bring down condensation. The magnetic rack was used to pellet the beads and collect the eluate. In addition, 15 µg of each whole cell lysate and flowthrough was combined with 1x SDS protein loading buffer, boiled for 10 min at 95°C, cooled on ice, and spun down.

The eluates were separated on a 9% SDS gel, and the whole cell lysates and flowthrough samples were separated on another 9% SDS gel. The two gels were transferred to nitrocellulose membranes, which were stained with Ponceau S and imaged, and then the membranes were destained in deionized water. The membranes were cut in half horizontally. The bottom halves were blocked in 3% BSA in 1x TBST, and the top halves were blocked in 5% milk in 1x TBST for 1 hr at room

temperature. The top halves were incubated face down on 500 µl of 1:3000 rabbit anti-RRBP1 (Abcam, catalog no. ab95983) in 1% milk in 1x TBST for 1 hr. The bottom halves were each rocked in 1:5000 mouse anti-V5 (Life Technologies, catalog no. R960-25) in 10 mL of 3% BSA in 1xTBST at room temperature for 1 hr. The membranes were then washed four times with 1x TBST for 5 min each time. Each membrane was rocked in 10 ml of either 1:2000 anti-rabbit HRP (Bio-rad) or 1:2000 anti-mouse HRP (Bio-rad) in 3% BSA in 1x TBST for 1 hr at room temperature. The membranes were washed 4 times for 5 min each time with 1x TBST and quickly rinsed four times with deionized water. They were developed with Clarity western ECL substrate (Bio-rad) for 5 min. The blots were then imaged on a UVP gel imager.

The data in this figure are representative of two independent experiments.

## Preparation of EM samples related to SYNJ2BP overexpression in wild type or RRBP1 knockout cells with or without protein translation inhibitors (related to *Figures 3B*, *4B and C*, *Figure 4—figure supplement 1B*, and *Figure 4—figure supplement 2*)

Wild-type HEK 293T cells or RRBP1 knockout cells were plated in 6-well plates pre-coated with 50 µg/ml human fibronectin such that there would be two wells of cells for each sample. Each well was transfected at 70% confluence with 8 µl of Lipofectamine 2000 in MEM and 2000 ng of pLX_TRC208-SYNJ2BP-V5, with 1000 ng of V5-APEX2-NLS serving as a co-transfection marker. For *Figure 4B* and *Figure 4—figure supplement 1B*, the wells were transfected with 1000 ng of Flag-APEX2-NLS as a co-transfection marker instead of V5-APEX2-NLS. The SYNJ2BP-V5 construct was omitted in the negative controls. The cells were transfected for 4 hr, and then the media was changed to complete cell culture media. For experiments with protein translation inhibitors, approximately 24 hr post-media change, the media in all of the wells was changed to 2 ml of complete cell culture media. For the samples that were treated with protein synthesis inhibitors, the media contained 200 µM cycloheximide or 200 µM puromycin. The plates were returned to the incubator (37°C, 5% $CO_2$) for 2 hr. If the experiment did not involve protein translation inhibitors, the samples proceeded directly to the fixation step about 24 hr post-media change.

The samples were rinsed and fixed with 2% glutaraldehyde (Electron Microscopy Sciences) in sodium cacodylate buffer (100 mM sodium cacodylate with 2 mM $CaCl_2$, pH 7.4), washed, quenched with glycine, and washed again as above in 'Characterization of ERM-APEX2 and APEX2-OMM by electron microscopy.' After the initial addition and removal of 2% glutaraldehyde in sodium cacodylate buffer, the samples were kept on ice until the overnight incubation in 2% aqueous uranyl acetate. The cells were overlaid with a solution of 1.4 mM DAB and 1 mM $H_2O_2$ for 30 min. This solution was then removed, and the samples were again washed five times in cold sodium cacodylate buffer for 2 min each time.

Each well was then overlaid with 1 ml of a 1:1 mixture of 2% $K_4[Fe(CN)_6] \cdot 3H_2O$ in sodium cacodylate buffer:2% $OsO_4$ (Electron Microscopy Sciences) solution in cold sodium cacodylate buffer for 2 hr. The samples were then washed five times in chilled Millipore water and then incubated in chilled 2% aqueous uranyl acetate (Electron Microscopy Sciences) at 4°C overnight.

The cells were brought to room temperature and rinsed with Millipore water. Then, the cells were scraped, resuspended in 50% ethanol in water, and pelleted at 700xg for 1 min. The supernatant was discarded, and the pellets were dehydrated as previously described (*Hung et al., 2014*) and infiltrated in EMBED-812 using 1:1 (v/v) resin and anhydrous ethanol for 1 hr, 2:1 resin:ethanol overnight, and then 100% resin overnight. The pellets then polymerized in fresh resin at 60°C for 48 hr. A diamond knife on a Leica Ultracut UCT was used to cut 50 nm sections, which were imaged at the Whitehead Institute Keck Microscopy Facility on a FEI Technai Spirit transmission electron microscope operated at 80 kV.

The data in *Figure 3B* are representative images from two independent experiments with ≥7 fields of view each. *Figure 4B* and *Figure 4—figure supplement 1B* are representative images from one experiment with ≥6 fields of view for each cell line. The images in *Figure 4C* and *Figure 4—figure supplement 2* are from one experiment with ≥6 fields of view for each condition.

## Quantification of mitochondria-ER contacts in EM images (related to *Figure 4D and E*)

Images from the samples in *Figure 4C* and *Figure 4—figure supplement 2* that were taken at 4800x magnification or higher were blinded and then imported into ImageJ. Images that contained wrinkles caused by imperfect sectioning were discarded. In cells depicting clear nuclear contrast as a result of the co-transfection APEX2-NLS marker, the mitochondrial perimeter for each clearly defined mitochondrion was outlined by drawing a freehand line and measured. Any segment of an ER tubule less than 80 nm from a mitochondrion was considered to be in contact with it. Rough ER was defined as an ER tubule with ribosomes along the visible segment of that tubule, regardless of which side of the tubule they were on. Smooth ER lacked ribosomes for the visible segment of that tubule. To measure the distance in between mitochondria and the ER, several evenly spaced measurements about 10–20 nm apart were taken along the length of the contact, with the first and last measurements marking the beginning and end of the contact. All of the measurements taken along this length were averaged to give the average distance in between the two organelles. These values were then averaged to find the average distance between the two organelles for a given sample. The mitochondrial perimeter demarcated by the first and last measurements was retraced with a freehand line to determine the length of the mitochondrion-ER contact.

For each sample, the fraction of mitochondrial surface in contact with ER was measured in two ways. First, we measured the fraction of mitochondrial surface in contact with ER for each individual mitochondrion by dividing the summed length of the mitochondrial surface in contact with the ER by the length of the perimeter of that mitochondrion. This analysis was repeated with mitochondria specifically for mitochondria-rough ER contacts and mitochondria-smooth ER contacts. Mitochondria that had a contact with an ER tubule that could not be confidently assigned as rough or smooth were not used for this analysis (*Figure 4D*).

The second metric calculated was the fraction of total mitochondrial surface in contact with the ER in a given sample. This was determined by summing the lengths of the mitochondrial surface in contact with ER in the whole sample and dividing it by the total mitochondrial perimeter (the sum of the lengths of all of the mitochondrial perimeters) present in the sample. This analysis was repeated for the fraction of mitochondrial surface in contact with rough or smooth ER (*Figure 4E*). As before, mitochondria that had a contact with an ER tubule that could not be confidently assigned as rough or smooth were not used.

## Assessment of mitochondrial and ER morphology in SYNJ2BP knockout cells (related to *Figure 4—figure supplement 3A*)

HeLa cells stably expressing Cas9 from a confluent T25 flask were trypsinized and resuspended in 10 ml of cell culture media. 250 µl of this cell suspension was plated per well in a 24-well plate. The next day, the cells were transfected with 500 ng of SYNJ2BP guide RNA (gRNA)-1 or SYNJ2BP gRNA-2 plasmid and 2 µl of Lipofectamine 2000 in MEM for 3 hr, after which the transfection solution was replaced with full cell culture media. Untransfected cells were processed in parallel. After another two days, each well from the 24-well plate was expanded into one well of a 12-well plate. After three days, the cells in each well were suspended in 1 ml of cell culture media, and 200 µl of each of these cell suspensions was plated on glass coverslips pre-coated with 50 µg/ml human fibronectin in DPBS in a 48-well plate. The next day, the cells were incubated for 30 min with pre-warmed 300 nM MitoTracker Deep Red (Invitrogen, catalog no. M22426) in full cell culture media at 37°C, 5% $CO_2$, washed three times with prewarmed DPBS, and then fixed with 3.7% paraformaldehyde in DPBS for 15 min on ice. The samples were then washed three times with DPBS and permeabilized with ice-cold methanol (−20°C) for 5 min. Then, the cells were washed three times with DPBS.

The samples were blocked with 3% BSA in DPBS for 1 hr at 4°C and then incubated plates with 1:100 rabbit anti-SYNJ2BP (Sigma Aldrich, catalog no. HPA000866) in 3% BSA in DPBS for 1 hr at 4°C. The cells were washed three times with 0.2% Tween in DPBS. Then, the plates were incubated with 1:500 anti-rabbit AF488 in 3% BSA in DPBS for 1 hr at 4°C. The samples were washed three times with 0.2% Tween in DPBS, incubated with 1:100 rabbit anti-PDI (Santa Cruz Biotechnology, catalog no. sc-20132) in 3% BSA in DPBS for 1 hr at 4°C, and washed again three times with 0.2% Tween in DPBS. Finally, the samples were incubated with 1:500 anti-rabbit AF405 in

3% BSA in DPBS for 1 hr at 4°C, washed three times with 0.2% Tween in DPBS, and imaged by confocal microscopy.

This experiment was performed once.

## Assessment of mitochondrial and ER morphology in RRBP1 knockout cells (related to *Figure 4—figure supplement 3B*)

HEK 293T cells were plated in a 6-well plate to generate lentiviruses carrying either RRBP1 gRNA-1 or RRBP1 gRNA-2. The next day, when the cells were ~70% confluent, the cells were transfected with 1000 ng of the guide RNA plasmid (see 'Generation of CRISPR knockout cell lines'), 100 ng of pVSV-G, and 900 ng of dr8.91 with 8 µl of Lipofectamine 2000 in MEM. After four hours, the transfection solution was removed and replaced with full cell culture media, and the plate was returned to the incubator.

Two days later, HeLa cells stably expressing Cas9 were plated in a 6-well plate. The next day, the supernatants containing the lentiviruses were filtered through 0.45 µm syringe filters, and 500 µl of each lentivirus was added to separate wells of stable Cas9 HeLa cells. Approximately 56 hr later, the transduced HeLa Cas9 cells were expanded into two T25 flasks each in media containing 8 µg/ml blasticidin and 0.5 µg/ml puromycin. The media was changed to fresh blastidicidin- and puromycin-containing media each day for 7 days (for the cells transduced with RRBP1 gRNA-1 lentivirus) or 8 days (for the cells transduced with RRBP1 gRNA-2 lentivirus) to select for the population of cells stably expressing both Cas9 and an RRBP1 gRNA.

These stable cells were then plated on glass coverslips pre-coated with 50 µg/ml human fibronectin in a 48-well plate. HeLa cells stably expressing Cas9 and a non-targeted gRNA (described in 'Generation of the stable cell lines for pull down of SYNJ2BP-V5 and immunoprecipitation-mass spectrometry') were plated in parallel. The next day, the cells were 60–80% confluent. Each well was transfected with 100 ng of pDisplay-mCherry-KDEL and 0.8 µl of Lipofectamine 2000 in MEM for four hours, after which the transfection solution was replaced with full growth media. Approximately 26 hr post-media change, the cells were incubated with 300 nM MitoTracker Deep Red in full growth media at 37°C, 5% $CO_2$. The cells were then washed three times with warm DPBS and fixed in 4% paraformaldehyde in fixation buffer for 15 min at room temperature. The samples were washed twice with 1x PBS and then stored at 4°C overnight. The next day, the samples were washed once more with 1x PBS. The cells were then permeabilized with 0.2% Triton X-100 in 1x PBS for 6 min at room temperature and then washed three times with 1x PBS. The samples were then blocked with 3% BSA in 1x PBS for 1 hr at room temperature and then rocked in 1:500 rabbit anti-RRBP1 antibody (Abcam, catalog no. 95983) in 1% BSA in 1x PBS for 1 hr at room temperature. The samples were washed four times for 5 min each time with 1x PBS, rocked in 1:1000 anti-rabbit AF488 in 1% BSA in 1x PBS for 35 min at room temperature, and washed again with 1x PBS four times for 5 min each time. The final wash was replaced with fresh 1x PBS, and the samples were imaged by confocal microscopy.

This experiment was performed once.

## Comparative cell growth assay (related to *Figure 4—figure supplement 3C*)

Five sterile, black, clear bottom, 96-well plates were pre-coated with 100 µl of 50 µg/ml human fibronectin in DPBS for at least 20 min at 37°C under 5% $CO_2$. The wells were washed twice with 100 µl of DPBS. Clonal cell lines of HEK 293T expressing non-targeted Cas9, SYNJ2BP knockout, or RRBP1 knockout were seeded in triplicate into wells of each plate at 2000 cells/well. An additional triplicate of coated wells without cells served as background subtraction in each plate. One plate was immediately assayed after plating for cell viability by the CellTiter-Glo Luminescnt Viability assay (Promega). Subsequent plates were assayed at 24 hr intervals for four consecutive days.

This experiment was performed once.

## Glucose/galactose cell viability assay (related to *Figure 4—figure supplement 3D*)

As calcium signaling at mitochondria-ER contacts has previously been associated with stimulation of mitochondrial tricarboxylic acid cycle dehydrogenases (*Rimessi et al., 2008*), we assayed our

knockout lines for defects in mitochondrial oxidative phosphorylation (OXPHOS). As an indirect measure of OXPHOS activity, we evaluated the comparative viability of knockout cells in media supplemented with glucose or galactose. Mammalian cells cultured in galactose are forced to rely on mitochondrial OXPHOS to generate ATP (*Reitzer et al., 1979*), causing cells with impaired OXPHOS to be at a growth disadvantage (*Gohil et al., 2010*; *Robinson et al., 1992*).

Two sterile, black, clear bottom, 96-well plates were pre-coated with 100 µl of 50 µg/ml human fibronectin and washed as in 'Comparative cell growth assay' above. Clonal cell lines of HEK 293T expressing non-targeted Cas9, SYNJ2BP knockout or RRBP1 knockout were seeded in triplicate into wells of each plate at 2000 cells/well. An additional triplicate of coated wells without cells served as background subtraction in each plate. About 24 hr after seeding, cells were washed with DPBS and the growth medium was replaced with media containing 10% FBS, 100 units/ml penicillin, 100 µg/ml streptomycin, and DMEM without glucose suppplemented with 10 mM glucose or 10 mM galactose. After 72 hr, cells were assayed for cell viability by the CellTiter-Glo Luminescent Viability assay.

This experiment was performed once.

### Click-iT assay (related to *Figure 4—figure supplement 3E*)

A sterile, black, clear bottom 96-well plate was pre-coated with 100 µl of 50 µg/ml human fibronectin in DPBS and washed as in 'Comparative cell growth assay' above. A clonal HEK 293T cell line expressing non-targeted Cas9, two clonal HEK 293T SYNJ2BP knockout cell lines, and two clonal HEK 293T RRBP1 knockout cell lines were plated at 10,000 cells/well in triplicate. One triplicate set of cells expressing non-targeted Cas9 was treated with 200 µM puromycin in full growth medium 1.5 hr prior to the addition of L-azidohomoalanine (AHA). Then, the media in all wells was changed to AHA medium (DMEM with 4.5 g/l glucose, no glutamine, methionine, or cysteine (Thermo Fisher Scientific) supplemented with 10% dialyzed FBS (Sigma), 4 mM L-glutamine (Thermo Fisher Scientific), 0.20 mM L-cysteine (Alfa Aesar), 100 units/ml penicillin, 100 µg/ml streptomycin, and 50 µM AHA) except for one triplicate set of cells expressing non-targeted Cas9, which served as an omit AHA negative control to measure AF488 alkyne non-specific sticking. The media for the puromycin-treated samples was replaced with AHA medium containing 200 µM puromycin. The plate was incubated at 37°C, 5% $CO_2$ for 35 min. The cells were then washed once with 1x PBS, pH 7.4 and then fixed with 4% paraformaldehyde in fixation buffer for 20 min at room temperature. The plate was kept under foil from this step until it was read in the platereader. All subsequent steps were performed according to the manufacturer's instructions for the Click-iT kit (Thermo Fisher Scientific). Measurements were recorded using a Tecan platereader.

To analyze the data, the AF488 signal was divided by the Hoechst 33342 signal in each well to approximate the amount of protein synthesis/cell. The values from each triplicate set of wells was averaged. The averaged AF488/Hoechst 33342 fluorescence intensity ratio of the omit AHA sample was subtracted from the corresponding value for each sample. These adjusted ratios were plotted in the bar graph displayed in *Figure 4—figure supplement 3E*. The error bars represent one sample standard deviation.

This experiment was performed once.

## Acknowledgements

We thank N Watson (Whitehead Institute Keck Microscopy Facility), T H Deerinck, and M H Ellisman (National Center for Microscopy and Imaging Research, University of California at San Diego) for performing electron microscopy. Y Sancak, A Markhard, C Amaya, and A Draycott assisted with the characterization of mitochondria-ER tether candidate knockout cell lines. T Hashimoto and S Calvo provided assistance with data analysis. We are grateful to P De Camilli, J Baskin, P Walter, T Rapoport, E L Snapp, and M Shoulders for helpful discussions. VH and SSL were supported by National Science Foundation Graduate Research Fellowships; SSL was also supported by a National Defense Science and Engineering Graduate Fellowship. Funding was provided by the National Institutes of Health (R01 CA186568 to AYT and R01 GM077465 to VKM) and the Howard Hughes Medical Institute Collaborative Innovation Award (to AYT and SAC). VKM is an Investigator of the Howard Hughes Medical Institute.

# Additional information

## Competing interests

Alice Y Ting: Listed as an inventor on a patent on peroxidase-mediated proteomic mapping technology (US Application No. 61/497155). The other authors declare that no competing interests exist.

## Funding

| Funder | Grant reference number | Author |
|---|---|---|
| National Institutes of Health | R01 CA186568 | Victoria Hung<br>Stephanie S Lam<br>Alice Y Ting |
| Howard Hughes Medical Institute | HHMI Collaborative Innovation Award | Victoria Hung<br>Stephanie S Lam<br>Namrata D Udeshi<br>Tanya Svinkina<br>Gaelen Guzman<br>Steven A Carr<br>Alice Y Ting |
| National Institutes of Health | R01 GM077465 | Vamsi Krishna Mootha |
| National Science Foundation | Graduate Research Fellowship | Victoria Hung<br>Stephanie S Lam |
| U.S. Department of Defense | National Defense Science and Engineering Graduate (NDSEG) Fellowship | Stephanie S Lam |
| Howard Hughes Medical Institute | Investigator | Vamsi Krishna Mootha |

The funders had no role in study design, data collection, and interpretation, or the decision to submit the work for publication.

## Author contributions

Victoria Hung, Stephanie S Lam, Conceptualization, Data curation, Formal analysis, Validation, Investigation, Visualization, Methodology, Writing—original draft, Writing—review and editing; Namrata D Udeshi, Formal analysis, Investigation, Methodology, Writing—original draft, Writing—review and editing; Tanya Svinkina, Investigation, Writing—original draft; Gaelen Guzman, Investigation, Gaelen Guzman processed all gel band samples for the OMM and ERM proteomic experiment, acquired the OMM and ERM proteome mass spectrometry data, and helped with starting the data processing for this experiment; Vamsi K Mootha, Formal analysis, Investigation, Writing—review and editing; Steven A Carr, Resources, Supervision, Funding acquisition, Investigation, Methodology, Writing—review and editing; Alice Y Ting, Conceptualization, Resources, Supervision, Funding acquisition, Investigation, Visualization, Methodology, Writing—original draft, Project administration, Writing—review and editing

## Author ORCIDs

Victoria Hung [iD] http://orcid.org/0000-0003-3972-2820
Stephanie S Lam [iD] http://orcid.org/0000-0002-2687-3470
Vamsi K Mootha [iD] http://orcid.org/0000-0001-9924-642X
Alice Y Ting [iD] http://orcid.org/0000-0002-8277-5226

## Decision letter and Author response

Decision letter https://doi.org/10.7554/eLife.24463.019
Author response https://doi.org/10.7554/eLife.24463.020

## Additional files

### Supplementary files

• Supplementary file 1. OMM and ERM proteomes. (a) OMM proteome: 137 proteins enriched by APEX2-OMM after filtering by the $\log_2(H/L)$ and $\log_2(H/M)$ SILAC ratios as described in Methods. (b) ERM proteome: 634 proteins enriched by ERM-APEX2 after filtering by the $\log_2(H/L)$ and $\log_2(H/M)$ SILAC ratios as described in Methods. Gray shading denotes parent and isoform-specific entries deriving from the same gene. (c) Mitochondrial orphans: List of 22 proteins from the OMM proteome (*Supplementary file 1a*) without prior mitochondrial annotation as defined in Methods. (d) Secretory pathway orphans: List of 72 proteins from the ERM proteome (*Supplementary file 1b*) without prior secretory annotation as defined in Methods. Gray shading denotes parent and isoform-specific entries deriving from the same gene. (e) OMMxERM cross list: List of 68 proteins that appear in both the OMM and ERM proteomes. Proteins are ranked by $\log_2(H/M)$ from Replicate 1 of the OMM proteomic experiment. (f) Proteins comparably labeled by APEX2-OMM and APEX2-NES: List of proteins from the OMM proteomic experiment that pass the $\log_2(H/L)$ cut-offs but do not pass the $\log_2(H/M)$ cut-offs. These proteins are strongly biotinylated by both APEX2-OMM and APEX2-NES and could be mitochondria/cytosol dual-localized proteins. (g) Proteins comparably labeled by ERM-APEX2 and APEX2-NES: List of proteins from the ERM proteomic experiment that pass the $\log_2(H/L)$ cut-offs but do not pass the $\log_2(H/M)$ cut-offs. These proteins are strongly biotinylated by both ERM-APEX2 and APEX2-NES and could be ERM/cytosol dual-localized proteins. (h) OMM proteomic data: Complete OMM proteomic data. All proteins with two or more quantified, unique peptides in either replicate are shown. (i) ERM proteomic data: Complete ERM proteomic data. All proteins with two or more quantified, unique peptides in either replicate are shown. (j) Column definitions: Definitions of the column headings for *Supplementary files 1a–1i*.

DOI: https://doi.org/10.7554/eLife.24463.013

• Supplementary file 2. Analysis of specificity and depth of coverage. (a) OMM true positive list: 79 established OMM-localized proteins used for calculation of OMM proteome coverage. Literature citation is provided for each entry. (b) ERM true positive list: 90 established ERM-localized proteins used for calculation of ERM proteome coverage. Literature citation is provided for each entry. (c) Sub-mitochondrial analysis: The sub-annotation of the set of proteins from the human proteome containing GO terms GO:0005741 for OMM, GO:0005758 for IMS, GO:0005743 for IMM, and GO:0005759 for mitochondrial matrix. Any protein with more than one sub-mitochondrial annotation was assigned to one compartment only according to this priority: OMM>IMS>IMM>mitochondrial matrix. Proteins detected in the OMM proteome are indicated in column I. (d) Sub-secretory analysis: The sub-annotation of the set of proteins from the human proteome containing GO terms GO:0005783 for endoplasmic reticulum, GO:0005794 for Golgi apparatus, and GO:0005886 for plasma membrane. Any protein with more than one sub-secretory annotation was assigned to one compartment only according to this priority: endoplasmic reticulum>Golgi apparatus>plasma membrane. Proteins detected in the ERM proteome are indicated in column H. (e) Soluble ER proteins: A list consisting of 132 proteins to check if ERM-APEX2 enriched any soluble ER lumen proteins. To generate this list, we searched for human proteins annotated with the GO term GO:0005788 for 'endoplasmic reticulum lumen' that also lack predicted transmembrane domains according to TMHMM and UniProt. Our ERM proteome contains 13 proteins, which are indicated in column E. (f) Cytosolic proteins: The set of proteins from the human proteome annotated with the GO term GO:0005829 for 'cytosol' that lack annotated or predicted transmembrane domains according to UniProt or TMHMM. Proteins detected in the ERM proteome are indicated in column E. (g) Column definitions: Definitions of the column headings for *Supplementary files 2a–2f*.

DOI: https://doi.org/10.7554/eLife.24463.014

• Supplementary file 3. Identification of SYNJ2BP binding partners. (a) SYNJ2BP-V5 IP-MS: Enriched proteins identified by mass spectrometry following immunoprecipitation of SYNJ2BP-V5 expressed in HeLa cells. The 56 listed proteins had two or more quantified, unique peptides; two or more 116/114 and 117/115 iTRAQ ratios; and Benjamini-Hochberg adjusted p-values<0.02 (moderated *t*-test). In addition, both the $\log_2(116/114)$ and $\log_2(117/115)$ values were >0. The 116/114 ratio reflects enrichment in the experimental sample compared to a negative control expressing a gRNA that does not target anything in the mammalian genome. The 117/115 ratio reflects enrichment in the

experimental sample compared to a negative control in which endogenous SYNJBP was knocked down, and SYNJ2BP-V5 was not expressed. (b) SYNJ2BP-V5 IP-MS data: Complete SYNJ2BP-V5 IP-MS data. All proteins with two or more quantified, unique peptides and two or more 116/114 and 117/115 iTRAQ ratios are shown. (c) Column definitions: Definitions of the column headings for *Supplementary files 3a–3b*.

DOI: https://doi.org/10.7554/eLife.24463.015

## Data availability

The following previously published datasets were used:

| Author(s) | Year | Dataset title | Dataset URL | Database and Identifier |
|---|---|---|---|---|
| Pagliarini DJ, Calvo SE, Chang B, Sheth SA, Vafai SB, Ong SE, Walford GA, Sugiana C, Boneh A, Chen WK, Hill DE, Vidal M, Evans JG, Thorburn DR, Carr SA, Mootha VK | 2008 | A mitochondrial protein compendium elucidates complex I disease biology | https://www.broadinstitute.org/mitocarta/archived-version-mitocarta10-inventory-mammalian-mitochondrial-genes | Broad Institute, archived-version-mitocarta10-inventory-mammalian-mitochondrial-genes |

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
