## [Decision Letter]

Thank you for submitting your article "Proteomic mapping of outer mitochondrial and ER membranes by proximity biotinylation and discovery of a tethering complex" for consideration by *eLife*. Your article has been favorably evaluated by Anna Akhmanova (Senior Editor) and three reviewers, one of whom, David Pagliarini (Reviewer #1), is a member of our Board of Reviewing Editors.

The reviewers have discussed the reviews with one another and the Reviewing Editor has drafted this decision to help you prepare a revised submission. Although we recognize that you have submitted this manuscript as a Tools and Resources paper, the consensus view is that your conclusions require some functional validation as detailed below.

Summary:

This manuscript by Hung et al. describes the use of a previously developed technique, proximity biotinylation, to discover novel proteins localized to cellular compartments with relative inaccessibility to standard experimental techniques including the outer mitochondrial membrane (OMM) and the endoplasmic reticulum membrane (ERM). Through purification of mitochondria from various tissues followed by mass spectrometry-based proteomics and computational approaches, and via previous APEX2-based studies by these authors, much of the mitochondrial proteome has been established and recorded in databases such as MitoCarta2.0. However, probing cytosol-facing proteins of the OMM and ERM presents a challenge due to the loss or depletion of outer membrane proteins during the purification process. Another issue lies in the poor overlap observed between proteomic MS datasets for ER extracts. Proximity biotinylation relies on probing cells using an engineered APEX2 peroxidase genetically modified to target a specific cellular region, a biotin-phenol (BP) substrate, and hydrogen peroxidase (H2O2). Upon the addition of H2O2 for 1 minute, APEX2 converts BP to BP radicals which can then covalently tag proteins in their immediate vicinity. Cells are lysed and the biotinylated proteins are enriched using streptavidin beads followed by MS analysis.

In the present manuscript, the authors construct modified APEX2 peroxidases with targeting signals for native OMM and ERM proteins, as well as a cytosolic control APEX2 containing only a nuclear export sequence. Localization of the constructs was verified via fluorescence microscopy. The authors used Stable Isotope Labelling with Amino acids in Cell culture (SILAC) to radiometrically quantify the relative proximity of each detected protein to their targeted location versus the cytosol. The experiments resulted in the detection of 137 potential OMM proteins, and 634 potential ERM proteins for a total of 94 potentially novel proteins between both locations. The authors chose to focus on closer inspection of the proteins involved in mitochondria-ER contact sites and found SYNJ2BP, a protein of unknown function, was enriched in both the OMM and ER proteomes. Overexpression of SYNJ2BP in HEK 293T cells resulted in increased mitochondrial contacts with the rough endoplasmic reticulum. To uncover the binding partner of SYNBP1, the authors utilized the CRISPR/Cas9 system to remove endogenous SYNBP1 and expressed V5-tagged SYNBP1 in Hela cells followed by immunoprecipitation with anti-V5 beads. The interacting proteins were analyzed through LC-MS/MS. Of the 56 discovered proteins, the authors reasoned that only RRBP1 contained the characteristics necessary to act as the ER binding partner for SYNBP1, presenting a potentially novel mammalian tethering complex between the mitochondria and ER.

Essential revisions:

The reviewers were all in agreement that this was elegant, technically sound investigation into a timely and important area of organellar communication. The reviewers commented on the particularly impressive specificity of the authors' approach, and the use of relevant controls for contact site enriched proteins. The narrative style, including extensive open discussion of the experimental limitations of the work, was also well received. Nonetheless, the reviewers agreed that the further evidence of direct interaction between SYNJ2BP and RRBP1 would greatly strengthen their conclusions, as would some evidence of the functional relevance of the tethering complex. In particular, the reviewers request that the authors address the following concerns.

1) While the overexpression data and localization of SYNJ2BP is convincing, the functional role of the protein within membrane contact sites remains underexplored, as demonstrated by the lack of any observed phenotype for CRISPR KO of SYNJ2BP or RRBP1 (as discussed by the authors). Does a phenotype exist in different cell lines having higher ER-OMM contact density as the authors suggest? In yeast, ERMES is important for mitochondrial lipid biosynthesis. Are there any changes in the lipidomes (whole cell or mitochondrial) of the KO lines? Is mitochondrial phosphatidylserine biosynthesis disrupted? Do any protein abundance or PTM changes exist in the KO lines that indicate a defect in mitochondrial or ER biology? Any changes in mitochondrial biogenesis or OxPhos activity? Of course, all of this issues would not need to be resolved, but some connection to a relevant phenotype would strengthen the authors' central thesis.

2) In their IP-MS experiments, the authors identified 56 interactors for SYNJ2BP. Given the presences of a PDZ domain on this protein, they then searched these 56 for proteins for targets possessing PDZ-binding motifs that might serve as a tethering partner for SYNJ2BP. This analysis led them to RRBP1, the only protein with such a motif. Given the central importance of this PDZ-PDZ-binding motif interaction (the reason why RRBP1 was chosen), the authors should test its importance for the SYNJ2BP overexpression phenotype. This should be doable given that the authors seem to have some of the cloning tools already in hand. How does a SYNJ2BP protein behave with a truncated PDZ domain? Is it possible to use a mapping approach to identify the interaction site between both putative tethers?

---

## [Author Response]

*Essential revisions:*

*The reviewers were all in agreement that this was elegant, technically sound investigation into a timely and important area of organellar communication. The reviewers commented on the particularly impressive specificity of the authors' approach, and the use of relevant controls for contact site enriched proteins. The narrative style, including extensive open discussion of the experimental limitations of the work, was also well received. Nonetheless, the reviewers agreed that the further evidence of direct interaction between SYNJ2BP and RRBP1 would greatly strengthen their conclusions, as would some evidence of the functional relevance of the tethering complex. In particular, the reviewers request that the authors address the following concerns.*

*1) While the overexpression data and localization of SYNJ2BP is convincing, the functional role of the protein within membrane contact sites remains underexplored, as demonstrated by the lack of any observed phenotype for CRISPR KO of SYNJ2BP or RRBP1 (as discussed by the authors). Does a phenotype exist in different cell lines having higher ER-OMM contact density as the authors suggest? In yeast, ERMES is important for mitochondrial lipid biosynthesis. Are there any changes in the lipidomes (whole cell or mitochondrial) of the KO lines? Is mitochondrial phosphatidylserine biosynthesis disrupted? Do any protein abundance or PTM changes exist in the KO lines that indicate a defect in mitochondrial or ER biology? Any changes in mitochondrial biogenesis or OxPhos activity? Of course, all of this issues would not need to be resolved, but some connection to a relevant phenotype would strengthen the authors' central thesis.*

We agree with the reviewers’ assessment that the functional role of SYNJ2BP remains unclear despite our best experimental attempts, and that an elucidation of function would greatly strengthen our story. This has been a difficult problem that we have worked on for the past two years (trying to obtain a phenotype upon KO). We already have a tremendous amount of negative data: we compared cell growth rates for SYNJ2BP and RRBP1 KO cell lines to wild-type cells and found no difference. To compare protein synthesis levels, we looked at labeling of the amino acid analog azidohomoalanine (AHA) using the Click-iT assay in SYNJ2BP or RRBP1 KOs. To assay mitochondrial OXPHOS activity, we evaluated comparative growth of KO cells in media supplemented with glucose (conditions favoring glycolysis) or galactose (favoring OXPHOS). We have now included these negative data in new Figure 4—figure supplement 3. From these efforts, we have a growing understanding of what biological processes are likely *not* related to the SYNJ2BP-RRBP1 interaction, which should be useful information for scientists who wish to follow up on our study.

We also would like to point out that it is unclear what known mitochondria-ER functions would be affected by the specific subtype of mitochondria-rough ER contacts we observe. Current understanding of the functions of mitochondria-associated membranes (MAMs) has not included any functions specific to mitochondria-rough ER contacts. Lipid shuttling, calcium signaling, and the control of OXPHOS associated with calcium signaling are all functions traditionally associated with mitochondria-smooth ER contact sites. Conventional wisdom has even delineated MAM distances to <30 nm, which is far smaller than the large distances we observe in our induced mitochondria-rough ER contacts. It is unclear that disruption of these mitochondria-rough ER contacts would even result in a detectable known phenotype. In light of these limitations, we believe that elucidation of function of such a heretofore uncharacterized organellar interaction will be far from straightforward, and indeed will require such a high level of resources and time as to warrant a separate project and publication. We have expanded our discussion of mitochondria-rough ER versus mitochondria-smooth ER contacts in the revised text.

*2) In their IP-MS experiments, the authors identified 56 interactors for SYNJ2BP. Given the presences of a PDZ domain on this protein, they then searched these 56 for proteins for targets possessing PDZ-binding motifs that might serve as a tethering partner for SYNJ2BP. This analysis led them to RRBP1, the only protein with such a motif. Given the central importance of this PDZ-PDZ-binding motif interaction (the reason why RRBP1 was chosen), the authors should test its importance for the SYNJ2BP overexpression phenotype. This should be doable given that the authors seem to have some of the cloning tools already in hand. How does a SYNJ2BP protein behave with a truncated PDZ domain? Is it possible to use a mapping approach to identify the interaction site between both putative tethers?*

We agree with the reviewers that a finer mapping of the SYNJ2BP-RRBP1 interaction would strengthen our work. We have already attempted these experiments but were stymied by technical difficulties. SYNJ2BP is a small, 145 amino acid protein, of which 88 residues form the cytosol-facing PDZ domain. Our previous attempts to truncate this protein with full or partial deletions resulted in severely impaired expression or targeting of this protein to mitochondria. Given that the PDZ domain of SYNJ2BP forms almost the entire cytosolic domain, we believe it is reasonable to infer that the interaction involves this domain. Likewise, RRBP1 is a protein that possesses up to 54 copies of a decapeptide repeat, which has rendered cloning and amplification of this gene product utterly intractable in our hands despite numerous attempts. This has prevented us from mutating the PDZ-binding domain and overexpressing RRBP1 to do the converse mapping experiment.